EMBO
Molecular Medicine

# γ-Secretase inhibitors in cancer clinical trials are pharmacologically and functionally distinct

Yong Ran[1,*], Fokhrul Hossain[2], Antonio Pannuti[2], Christian B Lessard[1], Gabriela Z Ladd[3], Joo In Jung[1], Lisa M Minter[4], Barbara A Osborne[4], Lucio Miele[2] & Todd E Golde[1,**]

## Abstract

γ-Secretase inhibitors (GSIs) are being actively repurposed as cancer therapeutics based on the premise that inhibition of NOTCH1 signaling in select cancers is therapeutic. Using novel assays to probe effects of GSIs against a broader panel of substrates, we demonstrate that clinical GSIs are pharmacologically distinct. GSIs show differential profiles of inhibition of the various NOTCH substrates, with some enhancing cleavage of other NOTCH substrates at concentrations where NOTCH1 cleavage is inhibited. Several GSIs are also potent inhibitors of select signal peptide peptidase (SPP/SPPL) family members. Extending these findings to mammosphere inhibition assays in triple-negative breast cancer lines, we establish that these GSIs have different functional effects. We also demonstrate that the processive γ-secretase cleavage pattern established for amyloid precursor protein (APP) occurs in multiple substrates and that potentiation of γ-secretase cleavage is attributable to a direct action of low concentrations of GSIs on γ-secretase. Such data definitively demonstrate that the clinical GSIs are not biological equivalents, and provide an important framework to evaluate results from ongoing and completed human trials with these compounds.

**Keywords** cancer therapy; NOTCH; γ-secretase inhibitors
**Subject Categories** Cancer; Neuroscience; Pharmacology & Drug Discovery

## Introduction

NOTCH signaling is an evolutionarily conserved pathway involved in cell fate control throughout development and postnatal self-renewing tissue differentiation. Dysfunction of NOTCH signaling has been directly linked to multiple human diseases, from developmental syndromes to adult onset diseases. Deregulated expression of NOTCH receptors, ligands, and targets, including overexpression and activation of NOTCH have been described in blood and solid tumors by several groups (Joutel *et al*, 1996; Oda *et al*, 1997; Hubmann *et al*, 2002; Jundt *et al*, 2002; McDaniell *et al*, 2006; Fabbri *et al*, 2011). NOTCH signaling promotes the self-renewal of cancer stem-like cells (CSC) in several malignancies and participates in tumor–stroma and tumor–endothelium interactions in CSC niches in primary and metastatic tumors (Pannuti *et al*, 2010; Gu *et al*, 2012; Takebe *et al*, 2015). There is now growing interest in developing therapies targeting NOTCH signaling pathway at different levels of the cascade. Monoclonal antibodies (Noguera-Troise *et al*, 2006; Ridgway *et al*, 2006; Hoey *et al*, 2009; Jenkins *et al*, 2012), antisense or RNA interference (Wang *et al*, 2006; Pietras *et al*, 2011), receptor decoys (Smas *et al*, 1997; Small *et al*, 2001; Funahashi *et al*, 2008; Kangsamaksin *et al*, 2015), and glycosylation/protease inhibitor strategies have been developed targeting NOTCH receptors and ligands (Panin *et al*, 1997; Bruckner *et al*, 2000; Moloney *et al*, 2000). Among them, γ-secretase inhibitors (GSIs) targeting receptor activation have been investigated in numerous preclinical models and a significant number of early clinical trials as anticancer agents (Espinoza *et al*, 2013).

NOTCH receptor activation is mediated by a series of proteolytic cleavages. NOTCH precursors are first processed by a furin-like convertase at the S1 site to generate NOTCH extracellular and NOTCH transmembrane-intracellular domain subunits in the trans-Golgi apparatus (Logeat *et al*, 1998). The two subunits form the mature NOTCH receptor through non-covalent interaction between the C-terminus of the extracellular subunit and the N-terminus of transmembrane-intracellular domain subunit (Blaumueller *et al*, 1997; Mumm *et al*, 2000). The mature receptors are trafficked to the plasma membrane; ligand binding leads to cleavage of NOTCH by a disintegrin and metalloproteinase domain-containing protein 10 or more rarely, 17 (ADAM10 or ADAM17) at the S2 site (Brou *et al*, 2000) located 10–15 amino acids before the transmembrane domain. The ectodomain shedding-like S2 cleavage releases a short extracellular peptide with unknown function and a short-lived intermediate transmembrane-intracellular domain (TMD-ICD), which becomes a

1   Department of Neuroscience, Center for Translational Research in Neurodegenerative Disease, and McKnight Brain Institute, College of Medicine, University of Florida, Gainesville, FL, USA
2   Department of Genetics and Stanley S. Scott Cancer Center, Louisiana State University Health Sciences Center, New Orleans, LA, USA
3   College of Pharmacy, University of Florida, Gainesville, FL, USA
4   Department of Veterinary and Animal Sciences and Program in Molecular and Cellular Biology, University of Massachusetts, Amherst, MA, USA
    *Corresponding author. Tel: +1 352 273 9458; Fax: +1 352 294 5060; E-mail: yran@ufl.edu
    **Corresponding author. Tel: +1 352 273 9458; Fax: +1 352 294 5060; E-mail: tgolde@ufl.edu

   

substrate for γ-secretase. γ-Secretase first cleaves TMD-ICD within the TMD near the inner leaflet (S3 site) and eventually releases the intracellular domain of NOTCH (NICD) (De Strooper *et al*, 1999; Saxena *et al*, 2001). In the canonical Notch signaling pathway, NICD is translocated to the nucleus and forms a NOTCH transcriptional complex with downstream partners. The peptide between the S2 and S3 sites is further processed at multiple S4 sites by γ-secretase in murine NOTCH1(Okochi *et al*, 2006). These later steps in cleavage are analogous to cleavage of the β-amyloid protein precursor that releases Aβ (reviewed in De Strooper *et al*, 2010).

γ-Secretase enzyme not only cleaves the amyloid precursor protein (APP) and NOTCH receptors but many other type I transmembrane proteins within their transmembrane domains. Presenilin 1 and presenilin 2 (PS1 and PS2) form the catalytic core of γ-secretase, and three accessory proteins, anterior pharynx-defective 1 (APH-1), nicastrin, and presenilin enhancer protein 2 (PEN2) are also required to complete the γ-secretase complex (De Strooper, 2003; Edbauer *et al*, 2003; Kimberly *et al*, 2003; Takasugi *et al*, 2003; Wolfe & Kopan, 2004). A number of GSIs have been developed that effectively inhibit γ-secretase cleavage in humans since γ-secretase was identified as a therapeutic target in Alzheimer's disease (AD) (Haass *et al*, 1992; Shoji *et al*, 1992). The prospect of long-term treatment of AD patients with current GSIs appears unfeasible due to side effects that limit the maximum tolerable dose to the point where the extent of long-term Aβ lowering is likely to be insufficient to have clinical impact (Cummings, 2010; Extance, 2010; Samson, 2010; Schor, 2011; Doody *et al*, 2013). However, because of its role in mediating signaling events from other proteins, with a primary focus on NOTCH1 signaling, γ-secretase has become a therapeutic target in cancer and immunologic diseases and may very well be a target in other conditions (Espinoza *et al*, 2013; Andersson & Lendahl, 2014).

A potential confounding factor in the repurposing of GSIs is the overwhelming focus on effects on APP and NOTCH1, ignoring effects on many other potential substrates. More than 90 γ-secretase substrates have been identified to date, including the four mammalian NOTCH receptors (NOTCH1-4), and the five canonical transmembrane NOTCH ligands (Haapasalo & Kovacs, 2011). It is not known whether γ-secretase cleavage of these other substrate contributes to anti-tumor activities *in vivo*. BMS-906024 is the only GSI for which there is published data showing equivalent effects on cleavage of NOTCH1-4 using transcriptional reporter assays (Gavai *et al*, 2015). Additionally, some preclinical GSIs cross-inhibit the signal peptide peptidase (SPP) and signal peptide peptidase-like proteins (SPPLs), which are a group of aspartic proteases that cleave type II membrane proteins (Grigorenko *et al*, 2002; Ponting *et al*, 2002; Weihofen *et al*, 2002). The SPP family is known to play a role in immune surveillance, signaling, virus maturation, and protein dislocation through substrate proteolysis regulation, though their potential function in cancer has not been elucidated (Weihofen *et al*, 2003; Nyborg *et al*, 2004, 2006; Parvanova *et al*, 2009). The effect of the clinical GSIs studied herein on SPP family members is not known.

In order to better characterize the clinical GSIs in trials for various cancers, we established novel *in vitro* γ-secretase activity assays for recombinant human NOTCH1-4, CD44, VEGFR1 as well as cell-based assays for NOTCH1-4 using chimeric Aβ-NOTCH1-4 proteins. We also tested these compounds in recently developed assays for the SPP/SPPL family members (Ran *et al*, 2015). We evaluated these GSIs for ability to inhibit NOTCH1 cleavage in two triple-negative breast cancer cell lines, MDA-MB-231 and MDA-MB-468, and for functional effects in secondary mammosphere formation assays in the cell lines. Collectively, these data demonstrate that the clinical GSIs are not pharmacologic or biological equivalents, and provide a framework to evaluate results from human trials with these compounds. These studies also provide further insight into the both mechanisms of (i) processive γ-secretase cleavage established for APP and (ii) the potentiation of substrate cleavage observed at low GSI concentrations.

# Results

## Cell-free γ-secretase assay revealed human NOTCH and other substrates were processed in similar pattern as APP

We prepared seven fusion protein substrates, $rNOTCH1_{sub}$, $rNOTCH2_{sub}$, $rNOTCH3_{sub}$, $rNOTCH4_{sub}$, $rVEGFR1_{sub}$, $rCD44_{sub}$, and $rAPPC100_{sub}$, for use in cell-free γ-secretase assays (Fig 1A). Figure 1B shows the purified recombinant substrates detected with anti-FLAG M2 antibody following SDS–PAGE. Although the substrates clearly form multimers as assessed by SDS–PAGE, further purification of the monomer did not improve cleavage efficiency except in the case of rAPPC100sub (Jung *et al*, 2014). Following incubation with CHAPSO-solubilized membrane, we detected cleavage of the recombinant substrates with IP followed by MALDI-TOF. $NH_2$-terminal, Aβ-like fragments (NTFs) were immunoprecipitated with anti-Aβ Ab5 (Fig 1C), and COOH-terminal fragments (CTFs) were immunoprecipitated with anti-FLAG M2 antibody (Fig 1D). Cleavage products produced in the absence (black line) or presence (red line) of a potent GSI, LY411575 (1 μM), enabled us to deduce which peptides were produced by γ-secretase cleavage and at which site in the recombinant substrate the cleavage was occurring (Figs 1C and D, and 2, and Table 1). For clarity, all cleavage sites were labeled according to the full-length NCBI reference proteins.

We were able to detect NTF produced by γ-secretase cleavage using substrates purified from inclusion bodies (IBs) for all substrates except rNOTCH4sub. We did successfully detect cleavage using rNTOCH4sub purified from the soluble lysate fraction. As summarized schematically in Fig 2, the γ-secretase cleavage of these substrates produces multiple NTF. More NTF ($n = 9$) were produced form rNOTCH1sub and rNOTCH2sub than other substrates. rVEGFR1sub had the fewest NTF generated ($n = 2$). All cleavage sites were within the predicted TMD of the substrate and were 6–19 residues from the positively charged residues that are predicted to be the membrane stop anchors.

Using these methods, we were able to detect the CTF produced from all substrates except rCD44sub. This enabled us to map the S3/ε-cleavage site that represents the initial site of substrate cleavage by γ-secretase. A single cleavage site was detected for rNOTCH1sub, rNOTCH2sub, rNOTCH4sub, and rVEGFR1sub. This contrasted to the multiple S3/ε-cleavage sites detected in rAPPC100-sub and rNOTCH3. Overall, these data suggest that the processive cleavage mechanism proposed to account for the heterogeneity of γ-secretase products from APP is qualitatively similar for the substrates studies here, but quantitatively distinct. Indeed, it

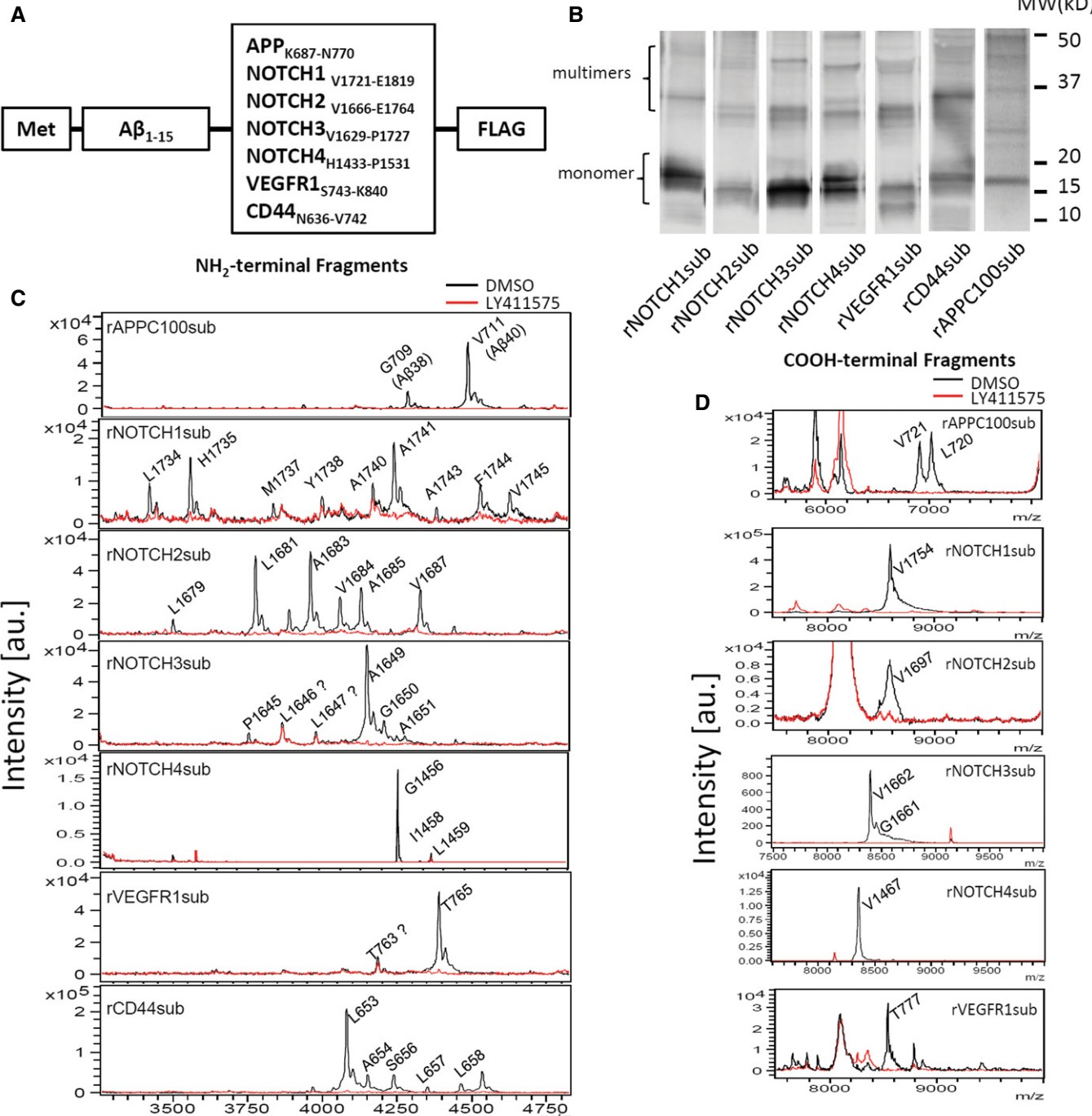

**Figure 1. The processive γ-secretase cleavage pattern of multiple substrates.**

A   Substrates based on Aβ peptide and γ-secretase substrate TMDs for cell-free assay.

B   Western blot of substrates overexpressed and purified from BL21.

C   Aβ-Nβ chimeric peptides (NTF) IP-MS of recombinant substrates after incubation with CHAPSO-solubilized CHO cell membrane with (red) or without (black) 1 μM LY411575.

D   COOH-terminal fragments (CTFs) IP-MS of recombinant substrates with (red) or without (black) 1 μM LY411575.

appears that the processive cleavage of the NOTCH1 and 2 substrates must be quite complex, and not limited to just tri- or tetrapeptide cleavages, in order to produce nine peptides following a single initial cleavage.

Although not suited for detailed dose–response studies due to challenges in precise quantification of the cleavage products, we nevertheless conducted several studies to examine how different GSIs affect cleavage in the cell-free assay. Structures of the GSI used

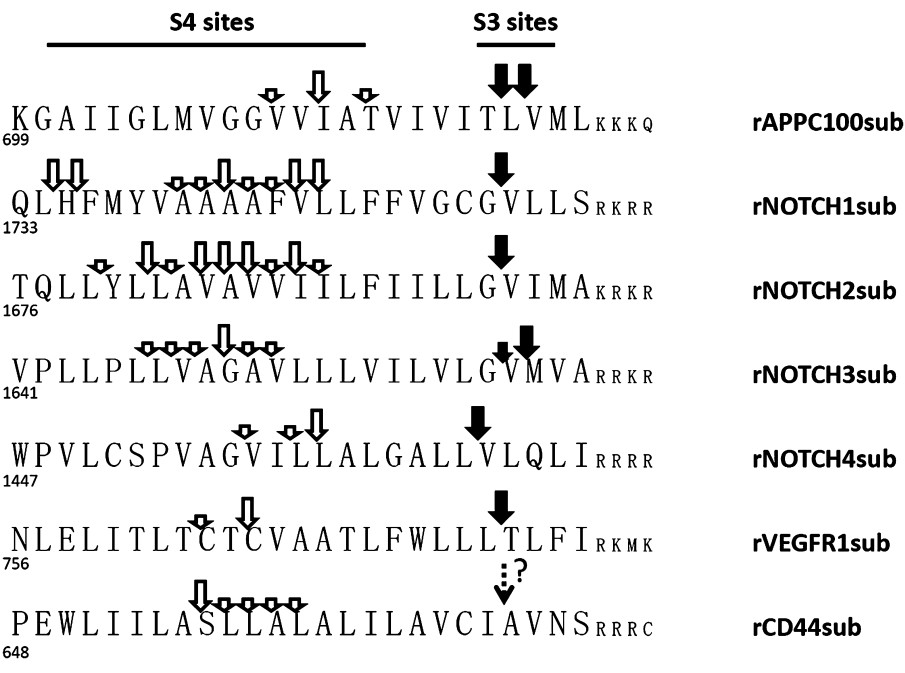

**Figure 2.  Putative substrate transmembrane domain and main cleavage sites observed in cell-free and cell-based assay.**
Solid arrows, S3 cleavage sites; open arrows, S4 cleavage sites; dashed arrow, predicted S4 site of CD44 (Okamoto *et al*, 2001; Lammich *et al*, 2002).

are shown in Fig 3A. 5 μM of these GSIs completely inhibited cleavage of rNOTCH1sub (Fig 3B). At 25 nM, BMS-906024 completely blocked cleavage of the rAPPC100sub and rNOTCH1-4subs (Fig 3C). We also evaluated a previously reported Notch-sparing GSI, Avagacestat (BMS-708163) (Gillman *et al*, 2010; Albright *et al*, 2013) using this assay (Appendix Fig S1). This compound appeared to inhibit APP and Notch cleavage equivalently, similar to what has been reported by others following the initial publications (Crump *et al*, 2012).

### Development of a cell-based γ-secretase cleavage assay for NOTCH substrates

In order to enable us to more definitively characterize the effects of various GSIs on different substrates, we generated H4 cells stably expressing cNOTCH1-4sub and cAPPC100sub (Fig 4A). Chimeric substrates were detected by WB using anti-FLAG antibody (Fig 4B). Multiple CTFs are detectable as well as potential dimers for all substrates. Multiple NTF are also detectable using anti-Aβ 6E10 antibody (Appendix Fig S2) for cNOTCH1, 3, and 4 substrates. Surprisingly, 1 μM DAPT treatment did not result in significant substrate accumulation except for cAPPC100sub.

The secreted Aβ-like peptides can be detected using Aβ ELISA method. Figure 4C shows the Aβ-like peptides in cNOTCH1sub, cNOTCH4sub, and cAPPC100sub stably overexpressing cell media increased to over 400 pM. cNOTCH2sub and cNOTCH3sub cells secreted over 100 pM Aβ-like peptides, which is about 10-fold greater than background levels. The secretion was almost completely inhibited when 1 μM GSI LY411575 was applied to cell cultures.

IP-MS revealed the cell-based assay generated similar cleavage products as the cell-free assay (Fig 4D and Table 1). Mouse Notch1 has two major S4 sites at A1731 and V1735 in cell-based assay

(Okochi *et al*, 2006). Here, we found the human NOTCH1 has three major sites at A1741, F1744, and V1745. Although Aβ-like peptides are detectable by ELISA, we were not able to detect cleavage product of cNOTCH4sub, cCD44sub, and cVEGFR1sub in the overexpressing cell lines using IP-MS. Notably, all cleavage products generated from the cell-based assay have two extra amino acid residues RG- at the NH₂-terminal, which arises from the NOTCH4 signal peptide, MDPPSLLLLLLLLLLLCVSVVRPRG at the expected signal peptide cleavage site.

### The clinical GSIs are not pharmacologic equivalents

Using these cell-based γ-secretase cleavage assays, we systematically investigated the substrate inhibition profile of BMS-906024, MK-0752, PF-308414, RO4929097, Semagacestat, and DAPT. We treated H4 cells stably expressing cNOTCH1-4sub with 4 pM to 10 μM of GSIs (Fig 5A and Table 2) and measured the Aβ-like peptide secreted into the media. These data show that BMS-906024 is a highly potent GSI and also the only compound to inhibit all NOTCH substrates nearly equivalently. Its IC₅₀s vary from 0.29 nM on cNOTCH2sub to 1.14 nM on cNOTCH3sub. PF-3084014 is also an extremely potent NOTCH inhibitor. It inhibits cNOTCH1, 3, and 4 substrates but shows remarkably potency for cNOTCH2sub with an IC₅₀ of 0.01 nM. RO4929097 shows a wide range of differential inhibition on each substrate; it inhibits cleavage of cNOTCH1sub with an IC₅₀ of 0.46 nM and cleavage of cNOTCH4sub with slightly lower potency, an IC₅₀ of 3.4 nM. However, RO4929097 at 0.5–2.5 nM significantly increases cleavage of the cNOTCH3sub, although at higher concentrations completely blocks cleavage from this substrate. This phenomenon, apparent potentiation of cleavage at low concentrations of inhibitor, has been observed for APP/Aβ in an animal model (Lanz *et al*, 2004, 2006; Siemers *et al*, 2005) and

**Table 1.  Molecular weight of Aβ-chimeric peptides and intracellular domain detected in cell-free assay and cell-based assay.**

| | | Cell-free assay | | Cell-based assay | |
|---|---|---|---|---|---|
| | Peptide | Observed m/z | Calculated MW | Observed m/z | Calculated MW |
| APPC100sub | CTF_L720[a] | 7016.8 | 7018.8 | – | – |
| | CTF_V721 | 6903.5 | 6905.7 | – | – |
| | A713[b] (Aβ42) | 4644.1 | 4645.3 | 4671.2 | 4670.2 |
| | V711 (Aβ40) | 4460.2 | 4461.1 | 4486.2 | 4486.0 |
| | G709 (Aβ38) | 4261.5 | 4262.8 | 4288.0 | 4287.7 |
| | G708 (Aβ37) | 4204.8 | 4205.7 | 4230.1 | 4230.7 |
| NOTCH1sub | CTF_V1754 | 8586.7 | 8588.5 | – | – |
| | F1748 | 5014.6 | 5013.6 | – | – |
| | V1745 | 4639.9 | 4640.1 | 5772.6 | 5771.4 |
| | F1744 | 4540.9 | 4541.0 | 5672.9 | 5672.3 |
| | A1743 | 4392.2 | 4393.8 | 5526.4 | 5525.1 |
| | A1741 | 4251.9 | 4251.7 | 5383.2 | 5382.9 |
| | A1740 | 4179.5 | 4180.6 | 5313.1 | 5311.8 |
| | V1739 | – | – | 5240.3 | 5240.8 |
| | Y1738 | 4009.4 | 4010.4 | – | – |
| | M1737 | 3847.7 | 3847.2 | – | – |
| | H1735 | 3567.6 | 3568.8 | – | – |
| | L1734 | 3430.2 | 3431.7 | – | – |
| NOTCH2sub | CTF_V1697 | 8573.2 | 8571.5 | – | – |
| | V1687 | 4341.0 | 4340.9 | 5473.5 | 5472.2 |
| | A1685 | 4142.1 | 4142.6 | 5274.5 | 5273.9 |
| | V1684 | 4074.4[c] | 4071.5 | 5203.3 | 5202.8 |
| | A1683 | 3972.0 | 3972.4 | 5103.9 | 5103.7 |
| | L1682 | 3900.2 | 3901.3 | – | – |
| | L1681 | 3788.1 | 3788.2 | – | – |
| | L1679 | 3511.7 | 3511.8 | – | – |
| NOTCH3sub | CTF_G1661 | 8452.5 | 8449.7 | – | – |
| | CTF_V1662 | 8395.4 | 8395.3 | – | – |
| | A1651 | 4289.9 | 4290.8 | – | – |
| | G1650 | 4219.1 | 4219.7 | – | – |
| | A1649 | 4161.0 | 4162.7 | 5295.8 | 5294.0 |
| | L1647 | 3991.7 | 3992.5 | – | – |
| | L1646 | 3878.5 | 3879.3 | – | – |
| | P1645 | 3765.4 | 3766.2 | 4898.0 | 4897.5 |
| NOTCH4sub | CTF_V1467 | 8357.5 | 8356.5 | – | – |
| | L1459 | 4683.9 | 4683.3 | – | – |
| | I1458 | 4570.1 | 4569.2 | – | – |
| | G1456 | 4358.1 | 4357.8 | – | – |
| CD44sub | A659 | 4548.5 | 4545.1 | – | – |
| | L658 | 4475.9 | 4474.0 | – | – |
| | L657 | 4361.4 | 4360.9 | – | – |
| | S656 | 4248.5 | 4247.7 | – | – |
| | A655 | 4159.7 | 4160.6 | – | – |
| | L654 | 4090.4 | 4089.6 | – | – |

**Table 1** (continued)

| | | Cell-free assay | | Cell-based assay | |
| --- | --- | --- | --- | --- | --- |
| | Peptide | Observed m/z | Calculated MW | Observed m/z | Calculated MW |
| VEGFR1sub | CTF_T777 | 8543.8 | 8545.7 | – | – |
| | T765 | 4399.0 | 4398.8 | – | – |
| | T763 | 4195.6 | 4194.6 | – | – |

[a]NH$_2$-terminal amino acid residue.
[b]COOH-terminal amino acid residue.
[c]Overlapped with a non-specific peak cause the value shift ~4 Da.

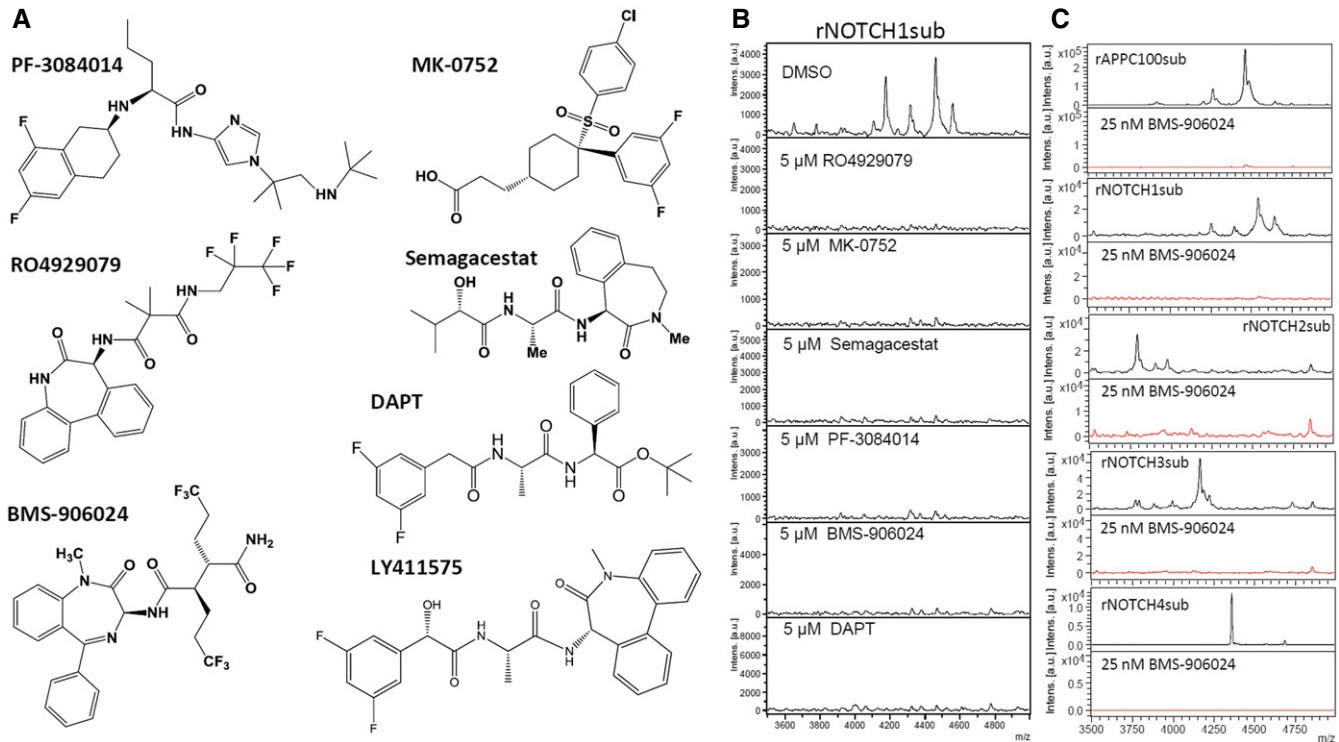

**Figure 3. GSIs in clinical trials for cancer therapy inhibit γ-secretase activity in cell-free assay.**

A  Chemical structure of GSIs used in this work.
B  5 μM GSIs completely abolished rNOTCH1sub γ-secretase cleavage.
C  IP-MS of NOTCH substrate with (red) or without (black) 25 nM BMS-906024.

cell cultures treated with select GSIs (Ran *et al*, 2014). Semagacestat and DAPT are also potent cNOTCH1sub inhibitors but potentiate both cNOTCH3sub and to a lesser degree cNOTCH4sub cleavage at lower concentrations. MK-0752 shows equivalent inhibition on cNOTCH1, 2, and 4 cleavages but is much less potent compared to BMS-906024, PF-3084014, and RO4929097. Again, it also increases cNOTCH3sub cleavage at low doses.

We next tested all GSIs on cAPPC100sub and full-length APP stably overexpressing cells. Similar to what we observed with cNOTCH substrates, BMS-906024 and PF-3084014 are the two most potent GSIs on cAPPC100sub and full-length APP. The low-dose potentiation of cleavage was observed for APP with RO4929097 and DAPT, and cAPPC100sub with RO4929097, Semagacestat, and MK-0752. However, the potentiation of cleavage was much higher with the cAPPC100sub, with up to twofold to threefold increases in

cleavage observed at certain concentrations (Fig 5B). Though no potentiation effect of BMS-906024 was initially observed, the effect emerged when we tested extremely low concentration of this GSI. When 1–4 pM BMS-906024 was used (Fig 5C), Aβ production from the cAPPC100sub was clearly increased ~1.5-fold. There was also a trend toward increased cleavage of cNOTCH1 and APP cleavage. The low-dose potentiation of γ-secretase cleavage was further mechanistically probed using *in vitro* assays with recombinant substrates and IP-MS. In these studies, we were able to document that BMS-906024 and Semagacestat-potentiated cleavage from rAPPC100$_{sub}$ at 0.03 and 0.16 nM, respectively (Fig 5D). Potentiated cleavage was also observed *in vitro* with rNOTCH3sub and rNOTCH4sub but with a biphasic response (Appendix Fig S3).

To extend the study to a more physiological system, we assessed endogenous NOTCH1 cleavage inhibition using the triple-negative

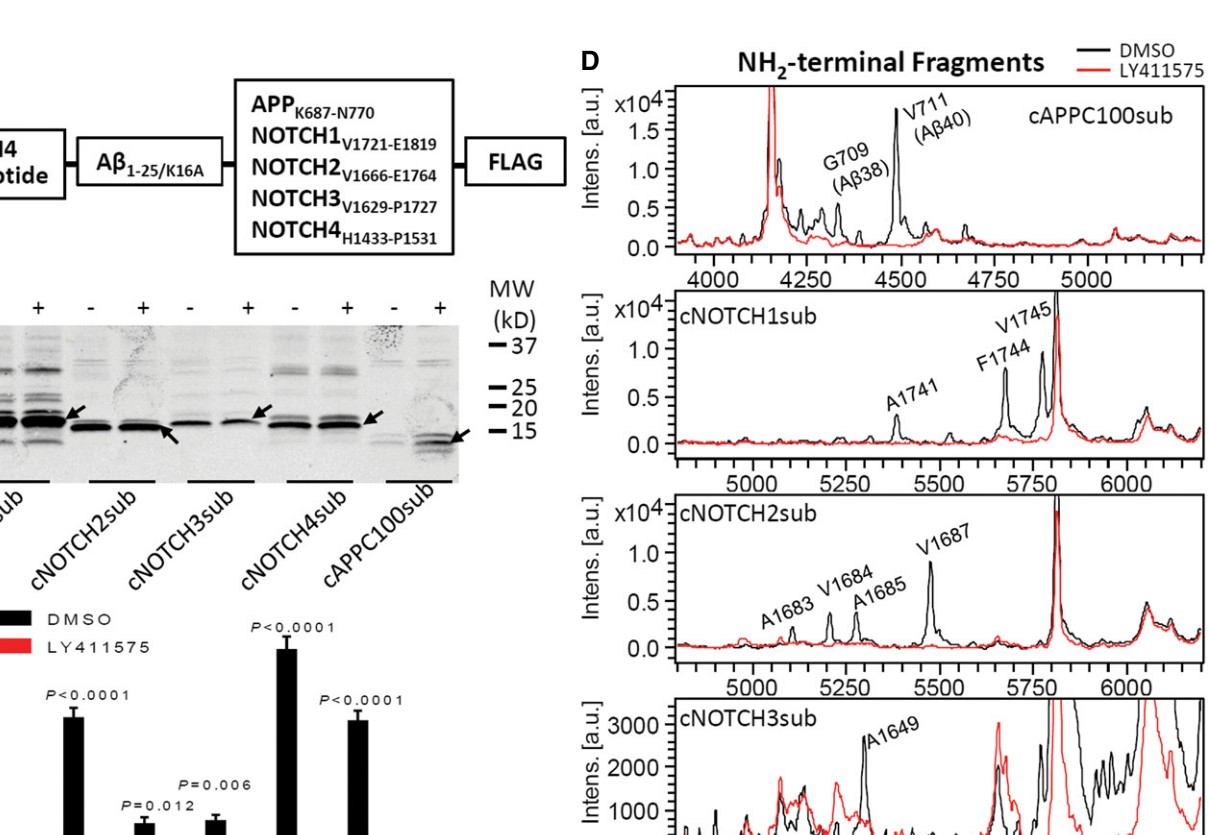

**Figure 4.  Chimeric NOTCH1-4 substrates are efficiently processed in cell-based assay.**

A   Substrates for cell-based assay.

B   Western blot of substrate transfected H4 cell lysate detected with 6E10 antibody. Arrows indicate substrate monomers.

C   Aβ was detected in culture media using Aβ ELISA, and the secretion was inhibited by GSI. Significance analysis with control (pCDNA3.1) was performed by one-way ANOVA using GraphPad. Values are mean ± SD of three tests.

D   Aβ-like peptides were detected in culture media by IP-MS (black). 1 μM LY411575 abolished all production (red).

breast cancer line MDA-MB-231 (Fig 6A). Full-length blots are shown in source data for Fig 6. NOTCH1 cleavage was detected by Western blotting using the Cell Signaling neo-epitope NICD antibody. In this cell line, all GSIs strongly decreased NICD levels at concentrations ranging from 100 nM to 12.5 μM. Semagacestat reduced NICD concentrations to near-baseline levels at all concentrations tested. BMS-906024, PF-3084014, and RO4929097 all reduced NICD to levels equal or lower than baseline at 100 nM. MK-0752 showed a clear dose dependence, reducing NICD to background levels or below at concentrations ≥ 0.5 μM. DAPT did not suppress NICD levels to baseline levels at any concentrations. Relative band intensities of the 100 nM GSI-treated samples (Fig 6B) suggest a potency rank that roughly matches what we got from previous cell-based assay (Table 2). Similar results were obtained in the same assay in MDA-MB-468 cells (data not shown). The inhibition of endogenous APP in MDA-MB-231 cells was also tested by detecting accumulation of APP COOH-terminal fragments (APP-CTFs). In the absence of GSI, APP-CTFs were not detectable or at very low level (Fig 6C). Starting from 20 nM, BMS-906024,

PF-3084014, or RO4929097 significantly inhibited APP-CTF process. Semagacestat, MK-0752, and DAPT started to inhibit APP-CTF process at 500 nM in MDA-MB-231 cells in a dose-dependent manner. For better quantification, we treated MDA-MB-231 cells with a single dose at 100 nM. Figure 6D summarized the APP-CTF band intensities normalized for DMSO group. The original blots are available in source data for Fig 6.

Suppression of breast cancer stem cells is thought to be a key mechanism of action for the anti-tumor activity of GSIs in breast cancer models and in patient-derived mammospheres (Takebe et al, 2015). Studies also suggest that primary mammospheres include stem-like and bulk (non-stem-like) cancer cells. Stem-like cells are enriched in secondary mammospheres, but their number declines after four to five passages (Dontu et al, 2003; Dey et al, 2009). Therefore, we compared the GSIs in a secondary mammosphere formation assay at 1, 5, and 10 μM concentrations (Fig 7A and B for 10 μM and Appendix Fig S4 for 1 and 5 μM). We separately counted large mammospheres (> 500 μm in diameter) and small mammospheres (100–500 μm in diameter). At 5 μM and especially at

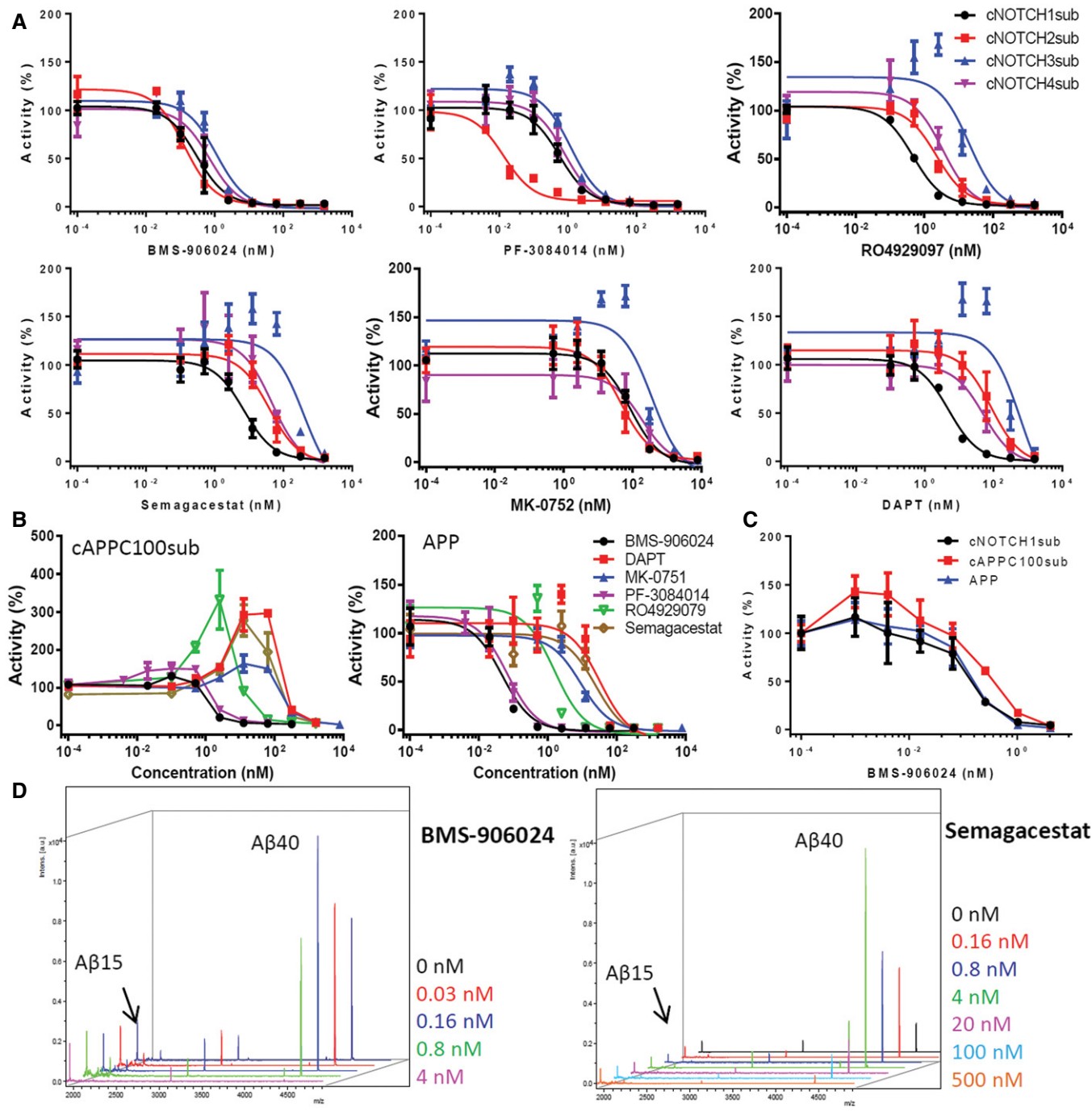

**Figure 5. The inhibition of GSIs on NOTCH substrates is not equivalent in cell-based assay.**

A IC$_{50}$ curves of BMS-906024, PF-3084014, RO4929097, Semagacestat, MK-0752, and DAPT on cNOTCHsub.

B Dose responses of cAPPC100sub and full-length APP cleavage to GSIs. The Aβ level of cells with DMSO was set as 100% activity.

C Low BMS-906024 dose potentiation of cNOTCH1sub, cAPPC100sub, and full-length APP.

D Low-dose potentiation of cleavage of rAPPC100sub was tested with *in vitro* assay and IP-MS. Aβ15 from rAPPC100sub was used as internal standards as shown on the left side of each spectrum.

Data information: All data were analyzed with GraphPad. All experiments were repeated three times. All data used for IC$_{50}$ curves are presented as mean ± SD.

10 μM, all GSIs decreased mammosphere-forming ability. However, there were no significant differences among the tested GSIs in all conditions in both cell lines. These data suggest that inhibition of

mammosphere-forming ability is common to all GSIs we tested. However, when we examined absolute cell counts, the PF-3084014 was significantly more potent than the other compounds in both cell

**Table 2.   IC$_{50}$ (nM) of GSIs on NOTCH substrates.**

| | cNOTCH1sub | | cNOTCH2sub | | cNOTCH3sub | | cNOTCH4sub | | Average[b] | cC100sub | APP | Reported IC$_{50}$s[e] | |
| --- | --- | --- | --- | --- | --- | --- | --- | --- | --- | --- | --- | --- | --- |
| | | | | | | | | | | | | Aβ40 | NOTCH |
| BMS-906024 | 0.29 | [1][d] | 0.14 | [2] | 1.14 | [1] | 0.71 | [1] | 0.57 | 1.46 | 0.05 | 1.6 | 0.7–3.4 |
| DAPT | 4.90 | [4] | 85.22 | [6] | 623.50 (265.2)[a] | [6] | 51.50 | [5] | 191.28 | 360.2 (212.5) | 32.68 | 20 | |
| MK-0752 | 87.44 | [6] | 46.62 | [5] | 370.00 (193.6) | [4] | 191.00 | [6] | 173.65 | 336.20 (154.3) | 9.03 | 5 | 55 |
| PF-3084014 | 0.6 | [3] | 0.01 | [1] | 1.21 | [2] | 0.81 | [2] | 0.66 | 2.466 (1.34) | 0.07 | 6.2 | 13.3 |
| RO4929097 | 0.46 | [2] | 2.24 | [3] | 19.8 (6.93) | [3] | 3.40 | [3] | 6.48 | 26.55 (23.97) | 1.41 | 14 | 5 |
| Semagacestat | 7.02 | [5] | 38.70 | [4] | 390.90 (128.8) | [5] | 45.98 | [4] | 120.65 | 175.00 (105.0) | 24.52 | 12.1 | 14.1 |
| Average[c] | 16.79 | | 29.82 | | 234.4 (98.01) | | 48.9 | | | 150.33 (83.09) | 11.29 | | |

[a]IC$_{50}$s in the squares are derived from curves excluding the Aβ level > 120%.
[b]Average IC$_{50}$s of each GSI on cNOTCH1-4sub in H4 cells.
[c]Average IC$_{50}$s of all GSIs on each cNOTCHsub.
[d]Numbers inside brackets [] represent the rank order of inhibition of each substrate by each compound.
[e]References are listed in Appendix Table S1.

lines we studied at 5 and 10 μM (Fig 7A and Appendix Fig S4). We also found that PF-3084014 decreased the total cell number and the CSC fraction in both cell lines under these conditions (Fig 7C), suggesting that it is not sparing "stem-like" cells. PF-3084014 caused a decrease in percentage of CD44$^+$CD24$^{low}$ cells at all concentrations tested, and a dose-dependent decrease in the absolute numbers of CD44$^+$CD24$^{low}$ cells.

**SPPLs are potential GSI targets**

It is known that SPP/SPPLs can be inhibited by select GSIs (Ran *et al*, 2015). Here, we tested the effect of various GSIs on the SPP/SPPL family members SPPL2a, SPPL2b, and SPP. We co-transfected SPPLs with the BRI2 based chimeric SPP/SPPL substrate that we have referred to as FBA into HEK 293t cells. SPPL-specific inhibitor

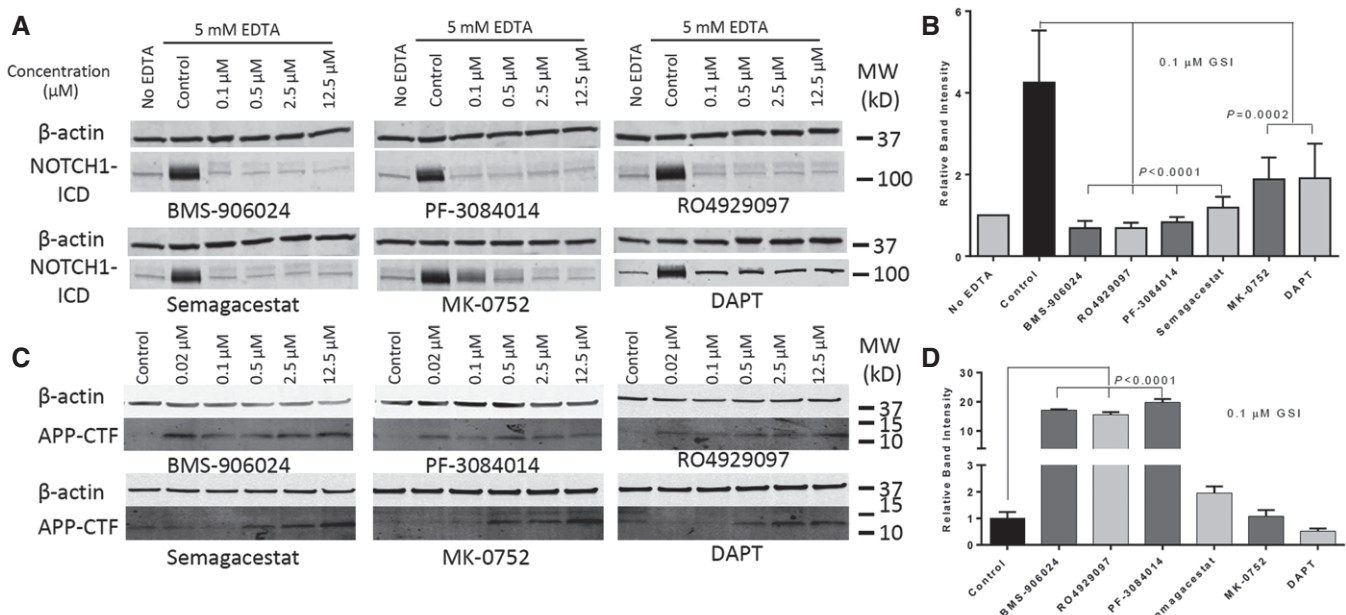

**Figure 6.   Endogenous NOTCH1 and APP-CTF cleavage inhibition by GSIs.**

A   Confluent MDA-MB-231 cells were treated with indicated concentrations of GSIs for 1 h and then with 5 mM EDTA for 5 min to induce NOTCH1 activation. The figure shows Western blot analysis of cell lysates treated with GSIs.

B   Band intensities of NOTCH1 ICD with 0.1 μM GSIs were normalized for β-actin using ImageJ Software. Values are mean ± SD of three tests.

C   APP-CTF Western blot of MDA-MB-231 cells treated with indicated concentration of GSIs for 16 h.

D   Band intensities of APP-CTF with 0.1 μM GSIs were normalized for DMSO control group. Values are mean ± SD of three tests.

Data information: Original blots are provided as source data. All band intensity data are derived from three independent experiments and are analyzed by one-way ANOVA using GraphPad Prism6 software. All experiments were repeated three times.
Source data are available online for this figure.

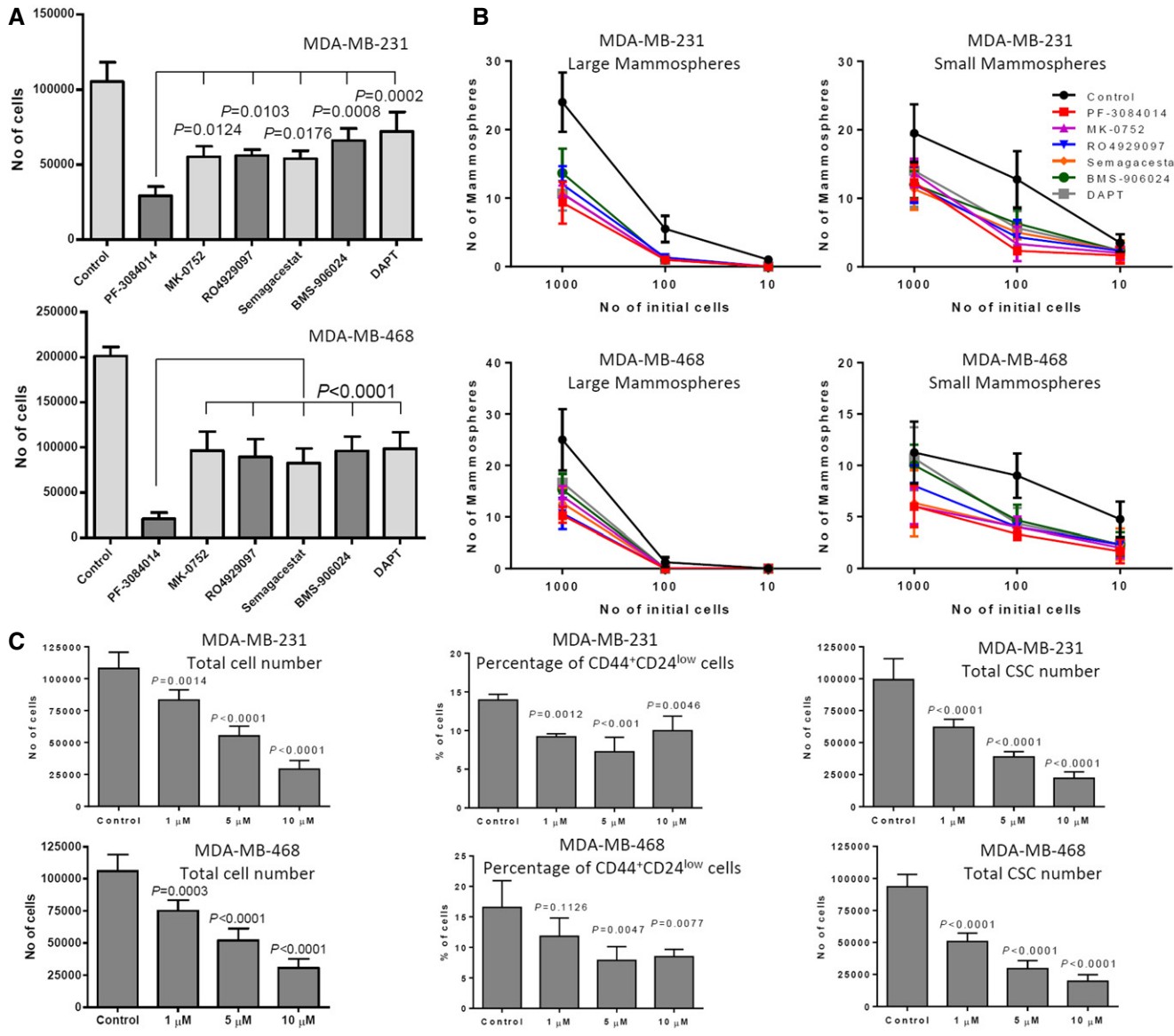

**Figure 7.  Mammospheres growth inhibition by different GSIs.**

A  Mammospheres from MDA-MB-231 and MDA-MB-468 cell lines were treated with 10 μM GSIs for 1 week. Absolute viable cell counts following GSI treatment are shown.

B  Viable cells (1,000, 100, and 10) from each GSI-treated conditions were then plated for limiting dilutions assay without further treatments.

C  Percentage and total number of cancer stem-like cells (CD44$^+$CD24$^{low}$) of PF-3084014-treated mammospheres were analyzed by flow cytometry.

Data information: All data were analyzed by one-way ANOVA using GraphPad Prism6 software. Values are mean ± SD of three tests.

(ZLL)$_2$-ketone was used as positive control. DAPT was not tested here as we and others have previously reported it had no effect on SPP activity at concentration up to 100 μM (Weihofen *et al*, 2003). In this system, 1 μM BMS-906024 and RO4929097 efficiently inhibited all SPPLs-catalyzed cleavage of FBA substrate (Fig 8A). PF-3084014 demonstrated mild inhibition at 1 μM. MK-0752 and Semagacestat did not show significant inhibition of SPPLs at 1 μM. Dose responses of BMS-906024, RO4929097, and PF-3084014 were performed with a similar strategy, but a stable SPPL2b/FBA overexpressing HEK 293 cell line was used instead of transient transfection (Fig 8B). Again BMS-906024 was found to be the most potent

SPPL2b inhibitor with IC$_{50}$ = 28.34 nM. RO4929097 has an IC$_{50}$ of 72.85 nM for SPPL2b. PF-3084014, a very potent GSI, showed only relatively mild inhibition over SPPL2b with IC$_{50}$ = 1,159 nM. Table 3 compares the GSIs' potencies on γ-secretase and SPPL2b. The average of IC$_{50}$s on cNOTCH1sub, cNOTCH2sub, cNOTCH4sub, and APP was used as the GSI's potency on γ-secretase.

To verify the finding in a non-overexpressing cell line, we applied the inhibitors to the mouse B lymphocyte cell line A20 (ATCC TIB-208). By analyzing the endogenous CD74 using In-1 antibody, we found 40 nM BMS-906024 or RO4929097 greatly stabilized the CD74 8 kDa NTF P8, a biomarker for SPPL2a cleavage, by

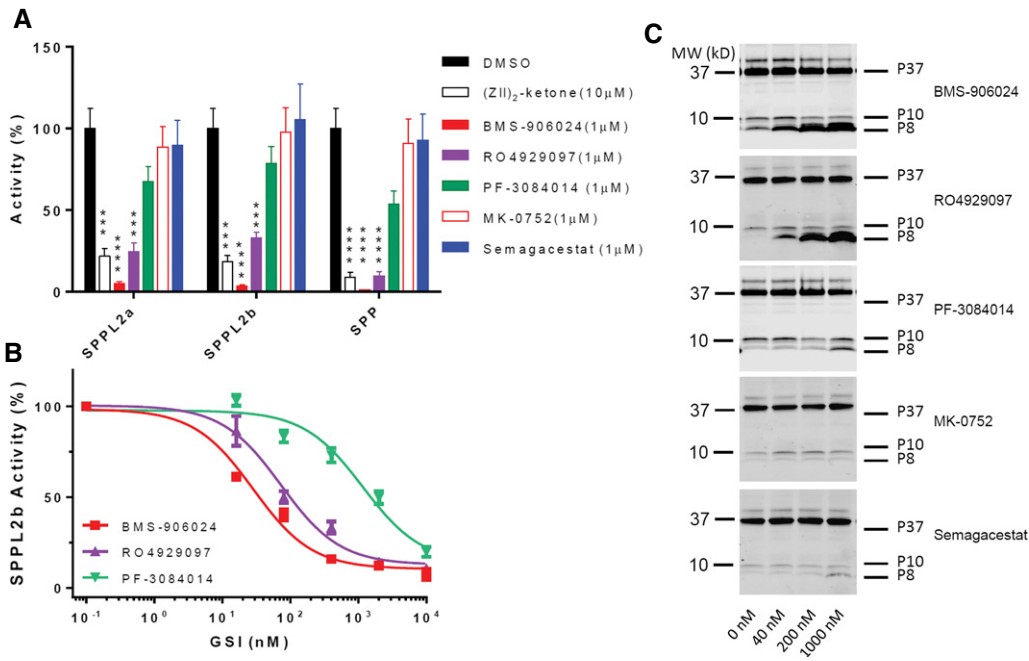

**Figure 8.  RO4929079 and BMS-906024 inhibit SPPLs activities efficiently.**

A  Conditioned media ELISA of HEK 293t cells co-transfected with FBA and SPPLs. The Aβ level of cells with DMSO was set as 100% activity. Values are mean ± SD of three tests. ***$P$ = 0.003; ****$P$ < 0.0001.

B  GSIs dose response of FBA/SPPL2b overexpression cells. Values are mean ± SD of three tests.

C  Turnover of endogenous CD74 P8 is inhibited by RO4929079 and BMS-906024. Cell lysate Western blot of A20 cells treated with GSIs was developed with In-1 antibody. MK-0752 and Semagacestat tests used the same control lane (0 nM).

Data information: Data were analyzed using GraphPad. Significance analysis with control (DMSO) was performed by one-way ANOVA.

Source data are available online for this figure.

---

**Table 3.  Clinical GSIs differentially inhibit SPPL2b (IC$_{50}$s in nM).**

|  | γ-secretase[a] | SPPL2b |
|---|---|---|
| BMS-906024 | 0.29 | 28.34 |
| RO4929097 | 1.88 | 72.85 |
| PF-3084014 | 0.37 | 1159.47 |

[a]The average of IC$_{50}$s on cNOTCH1sub, cNOTCH2sub, cNOTCH4sub, and APP.

impaired turnover (Beisner *et al*, 2013; Fig 8C). 40 nM PF-3084014 slightly stabilized P8, but did not improve by increasing concentration. Semagacestat only showed weak stabilization at 1,000 nM. MK-0752 did not show stabilization activity at the tested dose.

Overall, BMS-906024 and RO4929097 demonstrated the most robust inhibition without significant selectivity over different SPPL. The SPPL inhibition findings from the substrate overexpression method were also largely confirmed in the more physiologic conditions of the A20 cells.

## Discussion

We have used a suite of assays to more extensively evaluate the pharmacology and functional effects of multiple GSIs including four of the five compounds currently in clinical trials. The GSI, LY3039478, was not included in these studies, as it was only recently granted approval for human studies. Overall, our data

show that the clinical GSIs are both pharmacologically distinct with respect to their potencies for inhibiting APP and multiple NOTCH substrates in cell-based assays, and differ dramatically in their ability to kill mammosphere-forming cells in two molecularly distinct triple-negative cancer cell lines, in a way not clearly correlated to their ability to block NOTCH1 cleavage in the same cell line in two-dimensional cultures. Further, two of the clinical GSIs show nM IC$_{50}$s for inhibition of SPP/SPPL family members. This activity does not appear to correlate with mammosphere inhibitory activity. Collectively, these data indicate that the GSIs being tested in current trials for cancer are pharmacologically and likely functionally distinct. These distinctions may have important implications for understanding results from completed and ongoing trials.

An important and unexpected finding was the wide variance in the GSI potency against different NOTCH substrates (see Table 2). BMS-906024 was the only GSI that one would consider an equipotent inhibitor of all four NOTCH substrates and APP. PF-3084014 might also be considered a potent NOTCH inhibitor, but showed a remarkable > 50-fold higher potency for inhibition of a NOTCH2 substrate (IC$_{50}$ of 10 pM) and at low concentrations actually potentiates cleavage of NOTCH3. This compound showed the most potent mammosphere inhibitory activity in our assay. To our knowledge, only BMS-906024 was selected based on its ability to inhibit cleavage of all NOTCH isoforms (Gavai *et al*, 2015). Thus, these data indicate that a GSI must be empirically tested against multiple

substrates in order to determine their actual inhibition profiles, and that one cannot predict global substrate selectivity based on assessment of activity against one or two substrates. Further, our data suggest that selective substrate inhibition is typically the rule and not the exception with most GSIs. In the most extreme examples, GSIs can actually significantly potentiate γ-secretase cleavage of NOTCH3 or APP cleavage at concentrations where NOTCH1 cleavage is significantly inhibited. Development of substrate-selective γ-secretase cleavage inhibitors has been a long-standing goal, especially in the AD field, but previously reported Notch-sparing GSIs such Avagacestat, when tested in direct cleavage, assays often do not show substrate selectivity between NOTCH1 and APP (Crump *et al*, 2012). Nevertheless, as development of substrate-selective GSIs is of therapeutic interest, our data would suggest that with enough effort it might be possible to find inhibitors selective enough to achieve a desired biological effect against a specific substrate without significantly perturbing other substrates at clinically achievable doses. However, given the empirical nature of such an endeavor and the fact that there are well over 100 known γ-secretase cleavage substrates and likely several hundred unknown substrates, this would not be a trivial endeavor.

Another unexpected finding was that GSIs showed potentiation of γ-secretase cleavage at subinhibitory concentrations, and that this potentiation of activity appears to be a direct action on γ-secretase. The potentiation of cleavage had been well established for APP in previous studies but not reported for other substrates. The mechanisms for this effect have been debated, although some groups suggested it could be attributed to effects on endosomal degradation or to varying levels of substrate. Here, we present evidence that this potentiation appears to be (i) a direct action of the GSI on γ-secretase and (ii) a global phenomenon of GSIs, occurring at different subinhibitory concentrations for different substrates.

Over the last few years, the atomic structure of a γ-secretase complex (PSEN1, NCSTN, PEN-2, APH1A) has been revealed using single-particle cryo-electron microscopy (Bai *et al*, 2015). This elegant work suggests that there may be three detectable conformations of a single homogenous γ-secretase. Given that in human, four different complexes exist that variably contain PSEN1 or 2, or APH1A or B, one can envision a seemingly large number of conformers. Even further heterogeneity may be introduced by the lipid environment or even interaction with other proteins and substrates. Although speculative, we would suggest that the complex, substrate dependent, inhibitory profiles of various GSI on γ-secretase activity is likely due to a combination of select γ-secretase conformers which preferentially cleave certain substrates and differential binding of the GSI to various γ-secretase conformers. Indeed, one could envision that at a given concentration an inhibitor might bind to two different conformers and potentiate cleavage of one substrate in one γ-secretase conformer and inhibit cleavage of another substrate in another conformer.

The broader suite of γ-secretase assays we have developed provides important insight into the pharmacology of various GSIs, but there still remains a large gap in our knowledge regarding the functional consequences of a GSI with respect its potential as an anticancer therapy. Our data suggest that PF-3084014 has higher potency in mammosphere formation assays due to overall higher activity in both stem-like and non-stem-like cells in both the lines we tested. While all GSIs reduced mammosphere formation in

limiting dilution assays when identical numbers of viable cells were plated, PF-3084014 decreased the absolute numbers of viable cells in mammospheres more potently than the other compounds. Importantly, stem-like cells showed no relative resistance to this GSI, as they do to various cytotoxic chemotherapy agents. If anything, they appeared to be slightly more sensitive to this GSI than non-stem-like cells. However, our current characterization of these inhibitors does not provide clear insight into why PF-3084014 is the most potent GSI in the mammosphere inhibition assay.

Is this due to its enhanced potency for NOTCH2? An effect (either inhibitory or activating) on cleavage of another substrate? Its extracellular or intracellular stability? An off-target interaction? Clearly, such insights and, more importantly, insights into the potential clinical efficacy of the various GSI will require additional studies and comparison of pharmacological and biological properties of the GSIs with clinical data that emerges from completed and ongoing trials. Given the growing evidence that γ-secretase cleavage regulates signaling from numerous type 1 membrane proteins, the probability that extent of NOTCH1 inhibition is the only determinant of anti-tumor activity is small.

One of the challenges to translating preclinical data such as we have generated here to clinical studies is the lack of fluid-based or other *in vivo* biomarkers besides Aβ and a possibly APLP1-derived Aβ-like peptides (APL1b; Yanagida *et al*, 2009) to evaluate the effect of GSI on a given substrate in humans. Given the differential effects of many GSIs on various substrates, one would predict that looking at effects on Aβ levels as a measure of NOTCH inhibition could be misleading. By mapping the γ-secretase cleavage sites in multiple substrates using a relatively simple *in vitro* assays, we can now parlay this information to guide development of antibodies and ELISAs or even perhaps mass spectrometric assays that can be used to measure the levels of the Aβ-like peptides derived from a given substrate.

Besides informing on future biomarker development, mapping of the γ-secretase cleavage sites further confirms that the mechanism of γ-secretase cleavage is qualitatively similar for the many different substrates. An initial endopeptidase cleavage is likely followed by multiple, processive, carboxyl peptidase-like cleavages resulting in a spectrum of Aβ-like peptides with differing carboxyl termini. At present, it is hard to see how a consistent, processive cleavage model involving only tri- or tetrapeptides could account for the cleavage patterns we have observed in this study. Although this model has been invoked, and to some extent experimentally supported, with respect to cleavage of APP (Takami *et al*, 2009; Bolduc *et al*, 2016), other studies demonstrate that initial γ-secretase cleavage does not precisely define subsequent product lines (Ran *et al*, 2014) and that penta- and hexapeptides can be released during processing of APP by γ-secretase (Matsumura *et al*, 2014). Additional studies will be needed to understand how for example, a single initial endopeptidase cleavage of NOTCH1 and two substrates eventually produces nine Aβ-like peptides with no consistent spacing between the final cleavage sites. In these instances, a tripeptide stepwise cleavage model cannot account for the complexity of final products that are detected.

γ-Secretase has been proposed to be a therapeutic target in numerous human conditions (Golde *et al*, 2013). Further, well over a thousand studies using GSIs have been published. For the most part, these studies have assumed that GSIs, except for potency, are biologic equivalents. Our current data show that the biology of the GSIs is much more complex than previously appreciated, and the

general assumption of pharmacological and functional equivalency of GSIs is invalid. Indeed, our data provide an initial framework in which to evaluate current clinical GSIs being repurposed for cancer therapeutics and select GSIs used for preclinical studies. As data emerges from ongoing and completed clinical trials, these studies will help to inform the field regarding the properties of GSIs that are associated with clinical outcomes, and guide the development of a next generation of compounds.

# Materials and Methods

### Generation of recombinant substrates and cell-free γ-secretase cleavage assay

cDNAs encoding NOTCH1, NOTCH2, NOTCH3, NOTCH4, CD44, and VEGFR1 γ-secretase substrates were generated by gene synthesis conducted by Genscript (Piscataway, NJ, USA). The general design of the constructs was similar to a recombinant substrate (APP C100) that has been used by our group and others to assay Aβ production in in vitro γ-secretase assays. All constructs contain an $NH_2$-terminal amyloid β peptide (Aβ) epitope tag followed by the juxtamembrane region (JMD) of the given substrate and a FLAG tag (DYKDDDDK) at the COOH-terminal. A schematic representation of the substrates is provided in Fig 1A, and sequences of the cDNAs and the proteins are provided in Appendix Fig S5. For clarity, these substrates are referred to as recombinant substrates (e.g., Notch1 is $rNOTCH1_{sub}$). Substrate cDNAs were cloned into pET21 (Novagen, Billerica, MA, USA) for expression in bacterial cells.

pET21 plasmids containing the substrate cDNAs were transformed into BL21 E. coli. cells. Overexpression was induced with 0.5 mM isopropyl β-D-1-thiogalactopyranoside (IPTG). As pilot studies reveled that these substrates were primarily found in inclusion bodies (IBs), we focused on purification of the substrate form the IBs but cell lysate supernatant was also used to in some instances to purify substrates in soluble form. Cell pellets were collected and lysed with TBS Triton X-100 buffer (15 mM Tris–HCl, 100 mM NaCl, 0.5% Triton X-100). Raw IBs were separated by centrifugation at 3,000 g. The IBs were washed twice with wash buffer (15 mM Tris–HCl, 20 mM NaCl, 0.5% sodium deoxycholate) and low urea buffer (15 mM Tris–HCl, 20 mM NaCl, 1 M urea, pH 7.5) aided with a 3s microtip sonication followed by centrifugation at 3,000 g. The partially processed IBs were solubilized with urea buffer (15 mM Tris–HCl, 20 mM NaCl, 8 M urea, pH 7.5). Solubilized IBs were centrifuged at 18,000 g for 30 min at 4°C. With the exception of NOTCH4, all supernatants were purified with HiTrap Q columns (GE, Pittsburgh, PA, USA). NOTCH4 construct has a higher isoelectric point (7.2), so all buffers used in its purification were adjusted to pH 9.0.

CHAPSO-solubilized CHO cell membrane was prepared as described in previous report. 25 μg/ml of each substrate was incubated with the membrane (100 μg/ml total protein) in sodium citrate buffer (150 mM, pH 6.5, Roche Complete protease inhibitor added) for 2 h at 37°C. PF-3084014, RO4929079, MK-0752, Semagacestat (all purchased from MedChem Express, Monmouth Junction NJ), BMS-906024 (Maplewood, NJ), DAPT, and LY411575 (Sigma, St. Louis, MO) were tested at the range of 25 nM to 10 μM. The reaction was terminated by placing tubes on ice until immunoprecipitation.

### Cell-based γ-secretase cleavage assay

For the cell-based assay, all substrates contain a human NOTCH4 signal peptide sequence (MDPPSLLLLLLLLLLLLCVSVVRPRG), amyloid β peptide ($Aβ_{1–25/K16A}$) epitope tag followed by the juxtamembrane region (JMD) of the given substrate and a FLAG tag (DYKDDDDK) at the COOH-terminal. K16A mutant was created to decrease potential α-secretase cleavage. The cDNAs were generated by gene synthesis conducted by Genscript. A schematic representation of the substrates is provided in Fig 4A, and sequences of the cDNAs and the proteins are provided in Appendix Fig S6. For clarity, these substrates are referred to as cell-based assay substrates (e.g., NOTCH1 is $cNOTCH1_{sub}$). Substrate cDNAs were cloned into pCDNA3.1 (Invitrogen, Carlsbad, CA, USA) for expression in H4 cells (ATCC, Manassas, VA).

H4 cells were cultured in Opti-mem media (Thermo-Fisher) supplemented with Hyclone 6% fetal bovine serum (GE, Logan, Utah, USA) and 1% penicillin/streptomycin (Life Technologies, Grand Island, NY, USA). pCDNA3.1 plasmids containing the substrate cDNAs were transfected into H4 cells using FuGENE HD reagent (Madison, WI, USA). Stable cell lines were established using G418 selection. PF-3084014, RO4929079, MK-0752, BMS-906024, Semagacestat, DAPT, and LY411575 at the concentration of 4 pM to 10 μM were applied to cell cultures overnight. Cells and conditioned media were used for WB, IP, and ELISA.

Short-term endogenous NOTCH1 cleavage inhibition was tested in the triple-negative breast cancer cell lines MDA-MB-231 (ATCC, Manassas, VA). Cells were plated in 60 mm dishes and maintained as monolayer cultures in DMEM medium supplemented with 10% FBS, 1% penicillin/streptomycin, 1% L-glutamine (Life Technologies), and incubated at 37°C in a humidified 5% $CO_2$ incubator. After the cells reached ~75% confluency, cultures were pretreated with GSIs for 1 h in DMEM at 37°C in a 5% $CO_2$ incubator. GSIs were tested at concentrations of 0, 0.1, 0.5, 2.5, and 12.5 μM. Then, EDTA (5 mM, Life Technologies) was added to the cell cultures to induce Notch activation for 5 min. Following incubation, cell extracts were prepared by lysing cells in RIPA buffer (Santa Cruz Biotechnology) containing protease and phosphatase inhibitors. Western blotting was performed using Cleaved NOTCH1 (Val1744) antibody (Cell Signaling) with β-actin (Santa Cruz Biotechnology) as loading control. Band intensities were normalized for actin using ImageJ Software (NIH, USA).

### Mammospheres assays

For limiting dilution assay, we used two distinct TNBC cell lines. MDA-MB-231 is molecularly "Mesenchymal", PTEN wild type. MDA-MB-468 (ATCC, Manassas, VA) is molecularly "Basal-like" and PTEN null. GSI-treated (1, 5, and 10 μM twice/week, for 1 week) mammospheres were dissociated by trypsinization, and equal numbers of viable cells were re-plated at clonal density (1,000, 100, and 10 cells) without GSIs. After 7-day incubation, large mammospheres (> 500 μm) and small mammospheres (100–500 μm) were counted manually using a Nikon Eclipse Microscope. Next, we investigated whether PF-3084014 selectively affected cancer stem-like cells (CSCs) using both MDA-MB-231 and MDA-MB-468 cell lines. Mammospheres were treated with increasing concentration of PF-3084014 for 1 week, and CSCs ($CD44^+CD24^{low}$) (Azzam et al, 2013)

### The paper explained

**Problem**

γ-Secretase inhibitors (GSIs), originally developed as therapeutics for Alzheimer's disease based on their ability to reduce production of the amyloid β peptide, are being actively repurposed as cancer therapeutics based on the premise that inhibition of NOTCH1 signaling in select cancers is therapeutic. Over a thousand papers have been published using GSIs, and except for potency most of these have assumed that the GSIs are biological equivalents. However, this premise has not been systematically evaluated.

**Results**

Using novel assays to assess the effects of GSIs against a broader panel of substrates, we demonstrate that clinical GSIs are pharmacologically distinct. GSIs differentially inhibit various NOTCH substrates. Select GSIs even enhance cleavage of other NOTCH substrates at concentrations where NOTCH1 cleavage is inhibited. We establish that these GSIs have different functional effects in triple-negative breast cancer line mammosphere inhibition assays.

**Impact**

These data definitively demonstrate that the clinical GSIs are not biological equivalents and provide an important framework to evaluate results from ongoing and completed human trials with these compounds. These studies will help to inform the field regarding the properties of GSIs that are associated with clinical outcomes, and guide the development of a next generation of compounds.

were analyzed by flow cytometry using anti-human PE-CD44 (Clone G44-26) and FITC-CD24 (Clone ML5) (BD Biosciences) antibody.

### Cell-based SPPLs cleavage assay

SPPL2a, SPPL2b, SPP, and chimeric substrate FBA were generated as previously described (Ran *et al*, 2015). A20 cells (TIB-208; ATCC, Manassas, VA, USA) were cultured in RPMI-1640 medium (ATCC) supplied with 10% fetal bovine serum, 1% penicillin/streptomycin, and 0.05 mM 2-mercaptoethanol (Life Technologies). Various concentrations of GSIs and $(ZLL)_2$-ketone as indicated in figures were applied to transfected HEK 293t cells at 70–90% confluence or A20 cells at a density of $5 \times 10^5$ cells/ml with fresh media in 48-well format for 16 h. Cells and conditioned media were used for WB, IP, and ELISA.

### Immunoprecipitation and mass spectrometry

Immunoprecipitation and mass spectrometry of Aβ and Aβ-like peptides in cell-free assay or conditioned media were performed as previously described (Ran *et al*, 2014). Briefly, the peptides were immunoprecipitated using anti-Aβ Ab5 antibody bound to sheep anti-mouse IgG magnetic Dynabeads (Life Technologies) and eluted with 0.1% trifluoroacetic acid (TFA) in water. COOH-terminal fragments (CTFs) were immunoprecipitated with anti-FLAG M2 magnetic beads (Sigma). Eluted samples were mixed 2:1 with saturated α-cyano-4-hydroxycinnamic acid (CHCA) matrix (Sigma) in a mixture of acetonitrile (60%) and methanol (40%) and loaded onto a CHCA-pretreated MSP 96 target plate (Bruker, Billerica, MA). Samples were analyzed using a Bruker Microflex LRF-MALDI-TOF mass spectrometer.

### ELISA and Western blotting

Sandwich ELISAs used for Aβ detection were performed as previously described (Murphy *et al*, 1999; Ran *et al*, 2014). Briefly, Aβ and Aβ-like peptides in conditioned media were captured with Ab5 antibody and detected with horseradish peroxidase labeled mAb 4G8 (Biolegend). Synthetic Aβ1-40 was used as standard. All ELISAs were developed with TMB substrate (KPL, Gaithersburg, Maryland, USA). Bis-Tris precast gels (Biorad, Hercules, CA, USA) were used for all SDS–PAGE. Monoclonal anti-FLAG M2 antibody (Sigma) and Aβ1-16 antibody 6E10 (Covance) were used for Western blotting at 1:1,000 dilution. Anti-CD74 In-1 antibody (BD Bioscience, San Jose, CA, USA) was used for A20 cell lysate Western blotting at dilution of 1:1,000. MDA-MB-231 cell extracts were prepared by lysing cells in RIPA buffer containing protease and phosphatase inhibitors. The lysates were kept on ice for 30 min and vortexed every 10 min, and then centrifuged at 4°C for 10 min at 1,000 *g*. The supernatants were collected, and protein concentrations were determined by Bradford Assay. Pan-NOTCH1 antibody (C-20) (Santa Cruz Biotechnology) and cleaved NOTCH1 (Val1744) antibody (Developmental Studies Hybridoma Bank) were used at 1:500 and 1:100 dilution in the test of endogenous NOTCH1 inhibition with β-actin expression used as loading control. β-actin antibody was used at 1:1,000 dilution (Sigma). Band intensities normalized for actin using densitometry analysis with ImageJ Software (NIH, USA).

**Expanded View** for this article is available online.

### Acknowledgements

This study received funding from the NIH/NCI 1P01CA166009 01A1 (BAO, LM, TEG) and 2P01 AG020206 (TEG).

### Author contributions

TEG, LM, BAO, and YR designed research and analyzed data; YR performed most of the research and data analysis; FH and AP performed short-term endogenous NOTCH1 cleavage and mammospheres assays; CBL, GZL, and JIJ performed part of the *in vitro* assay; LMM edited the manuscript; YR, TEG, LM, and BAO wrote and edited the article.

### Conflict of interest

The authors declare that they have no conflict of interest.

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
