## [Review Process File · EMBO Molecular Medicine]

Manuscript EMM-2016-07265

γ -Secretase Inhibitors in Cancer Clinical Trials Are Pharmacologically and Functionally Distinct

Yong Ran, Fokhrul Hossain, Antonio Pannuti, Christian B Lessard, Gabriela Z Ladd, Joo In Jung, Lisa M Minter, Barbara A. Osborne, Lucio Miele, Todd E Golde

Corresponding author: Yong Ran & Todd Golde, University of Florida

Review timeline:

Submission date:	27 October 2016
Editorial Decision:	12 December 2016
Revision received:	10 March 2017
Editorial Decision:	11 April 2017
Revision received:	24 April 2017
Accepted:	25 April 2017

Editor: Céline Carret

Transaction Report:

1st Editorial Decision

12 December 2016

Thank you for the submission of your manuscript to EMBO Molecular Medicine. We have now heard back from the three referees whom we asked to evaluate your manuscript. Although the referees find the study to be of potential interest, they also raise a number of significant concerns that must be addressed in the next final version of the article.

You will see that while all referees find some merits in the study, referee 1 questions the cancer - experiments and this aspect of the work must be strengthened with more cell lines to be tested and mammosphere experiments to be performed as suggested; referee 2 finds the overall advance limited but agreed with referee 1 that focusing on the Notch cleavage inhibition in cancer cells as indicated would improve the study in terms of robustness and novelty; referee 3 is rather enthusiastic and only comments on minor issues. One common denominator however is the poor citation accuracy and this should be changed and improved throughout the article.

Given the balance of these evaluations, we feel that we can consider a revision of your manuscript if you can address the issues that have been raised within the space and time constraints outlined below. Please note that EMBO Molecular Medicine encourages a single round of revision and that, as acceptance or rejection of the manuscript will depend on another round of review, your responses should be as complete as possible.

Please read below for important editorial formatting for submitting the revised article (1 figure/file, format of reference list for example and other issues).

I look forward to receiving your revised manuscript.

***** Reviewer's comments *****

Referee #1 (Comments on Novelty/Model System):

Most of the technical data is sound but the cell biology assays for 'cancer stem cells' presented in figure 7 is technically and conceptually unsound and not fit for purpose.

Referee #1 (Remarks):

The paper is a much needed study of the diversity of gamma secretase inhibitor (GSI) action, with a particular focus on the inhibition of Notch receptor cleavage as a target in cancer. The details of the analyses of cleavage site preferences for individual gamma secretase targets are fascinating. In addition, the testing of the major chemical gamma secretase inhibitors developed the pharmaceutical industry reveals distinct patterns of substrate specificity that impacts selectivity of Notch receptor inhibition. This is important and has relevance and impact for the potential of these in cancer treatment.

However, the investigation falters when it comes to testing of the cellular biological impact of these different specificities for cancer. The use of a single breast cancer cell line limits the investigation and the assay that purports to test an effect on cancer stem cells is not fit for purpose since it does not test colony formation or self-renewal. Many publications have described this assay and the key steps are to plate at clonal density, use the inhibitor at time 0 to test effects on stem/progenitor cells and replat without inhibitor to test for an effect on self-renewal. Such methods have been previously described in detail in the literature. None of the above methods are followed and the assay used where inhibitor is added after secondary plating is most likely to test for an effect on proliferation. Consequently, they observe little effect and little difference between inhibitors. This part would have to be strengthened considerably with use of additional cell lines and preferably an *in vivo* limiting dilution assay to have any confidence that an effect on cancer stem cells is being seen.

Additional comments:

- 1) In the introduction and elsewhere, the citation of literature is limited too often to papers by the authors without reference to other relevant papers. For example, reference to decoys and antibodies for therapy fails to cite high impact papers (Eg. Kangsamaksin et al Cancer Discovery, 2015; Hoey et al., Cell Stem Cell, 2009)
- 2) Many acronyms are not defined on first use. Eg. APP, IB, CTP and NTP in abstract, results and figures.
- 3) Fig. 4: For B, it is not clear which band the arrows are pointing to. In 4C and D, LY411575 is used but no structure is given in fig. 3 and it is not defined whether it is a GSI or not.
- 4) Fig. 5: the line colours should be carefully chosen so that they can be distinguished.
- 5) Page 8, lines 10-17: The effects shown in Figure 5 are very interesting and discussed here where an increase in cleavage at low doses of GSIs are seen, particularly affecting Notch3 and 4. This could explain the differential effects of DAPT and other GSIs on Notch1 versus Notch3/4 that have been previously published.
- 6) Fig. 6: In A, the WB should indicate the band as N1-ICD. In B, the baseline control band intensity should be included. Error bars are shown but it is not clear from the legend whether these are from replicate WBs or not. Statistics and number of replicates should be included.
- 7) I am unsure about the relevance of SPPLs as targets to the main purpose of the paper. It seems to be an add-on. Is it relevant?
- 8) The relevance of the mammosphere assay to cancer stem cells and effects of Notch inhibition should be properly discussed if suitable data addressing this are included.

Referee #2 (Remarks):

In the present manuscript, Ran et al have investigated a variety of gamma-secretase inhibitors (GSIs) used in cancer clinical trials for their potencies and selectivity in inhibition of Notch and a few other substrates. As altered Notch signaling has been implicated in cancer, the focus of the study lies on the effects of the GSIs on the Notch1-4 substrates. In addition, the authors have investigated whether and how the selected GSIs would inhibit the gamma-secretase related SPP/SPPL proteases.

By identifying the gamma and epsilon cleavage sites in cell-free and cell-based assays, the authors show that Notch1-4, VEGFR and CD44 are processed in a similar way as APP. The authors further find that the GSIs tested can show differential effects on substrate cleavage and on cross-inhibition of SPP/SPPLs, such that the overall conclusion of the study is that the GSIs in cancer clinical trials are pharmacologically and functionally distinct.

As outlined below, I am afraid to say that the findings, conclusions and the concepts derived are, however, not really novel. It has been known since a long time that also other gamma-secretase substrates likely undergo sequential cleavages as APP by the identification of cleavage sites in the N-terminal to epsilon/site3 in a number of substrates (Notch1, CD44, APLP1, Neuregulin1). The authors confirm this concept by showing internal cleavages sites at gamma/site4 positions now also for Notch2, 3, and 4 and VEGFR. An interesting aspect is that some inhibitors seem to increase activity of processing at low-dose inhibition (e.g. for Notch3), a phenomenon which has already been known from APP processing.

In addition, that GSIs, which are not directed against the active such as the early transition-state analog inhibitors, which block cleavage of all substrates, can show differential effects on substrate cleavage with respect to inhibitor potency, is likely and has been shown earlier. Unlike the authors state in this study, I think it is commonly accepted in the field that GSIs are not necessarily considered biological equivalents except those that directly target the active site. Other GSIs that not directly target the active site (and several if not all of the GSIs studied here likely fall into this category) can as mentioned above of course show a certain degree of substrate specificity. In fact, this concept has been the basis for the development of Notch-sparing GSIs for Alzheimer's disease therapy. Differential effects of GSIs and/or complex-specific inhibitors have been reported for PS1 and PS2 gamma-secretases by others, as well as for SPP/SPPLs by the authors earlier (Ran et al. PlosOne 2015).

Some inhibitors have slightly different potencies on Notch1 processing in breast cancer cell lines and some inhibitors inhibit mammosphere formation. Similar findings on mammosphere formation have already been reported for two other GSIs previously (Grudzien et al. Anticancer Res. 2010). These effects are interesting and working out to which substrates these relate could provide important novel insights. The implications and significance for cancer therapy, which these findings may have, do, however, not yet really become clear from this study.

Lastly, the technical quality of the data is overall very good, although some conclusions made also require further experimental substantiation. However, as said above, the manuscript provides only little advance over previous studies and the overall impact might therefore be only moderate.

Specific points:

Results:

- The cleavage sites and relative site usage (peak height) does not always match between both cell-free (Fig. 1C, e.g. for Notch1: A1741 is major peak) and cell-based (Fig. 4D, e.g. for Notch1: A1741 is minor peak) assays and also not with previous published studies (e.g. CD44 major gamma-cleavage site between LA and LA as reported by Lammich et al. JBC 2002, but in this study more N-terminal between A and S). These discrepancies may relate to different construct designs with respect to ectodomain length and N-terminal epitope-tag usage but would deserve a critical discussion.
- The lack of substrate accumulation shown in Fig. 4B is indeed very surprising. Is this also observed with other potent GSIs such as L-685,458? In clarifying this issue, the authors should also check CTF accumulation of a suitable endogenous substrate to assess whether the observed lack of substrate accumulation in the presence of GSIs may be due to substrate saturation upon

overexpression. The immunoblots shown in Fig. 4B and Appendix Figure S4 are identical and not probed with different antibodies (Flag vs 6E10) as described (same main and background bands, same dirt, spots, etc.; please check carefully).

- For Figure 5A-D, it would be desirable that these data are further substantiated by immunoblot analysis of NTF (Abeta-like peptide) and CTF (ICD) cleavage products to see the increase of activity at low-dose inhibition also in this experimental setting. This would probably best be done in the cell-free assay. With respect to the mass-spectrometry analysis shown in Fig. 5D for Notch1, in the accompanying Appendix Fig. 5B, the signals (peak heights) for Notch3 and Notch4 unexpectedly and strangely first go down, then up again and then down again. Again, immunoblot analysis of cleavage products would be needed to make these data more robust.
- In Fig. 6, longer exposures comparable to that for BMS-906024 should be shown for the underexposed blots. Again, it appears that DAPT has not really been working (compare control 1 and lanes 4-6).

Literature:

- The primary references for the finding that gamma-secretase is a complex composed of presenilin, nicastrin, APH-1 and PEN-2 are incomplete and should also include Edbauer et al. *NatCellBiol* 2003 and Kimberly et al. *PNAS* 2003.
- The literature cited on the failure of GSI in AD clinical trials is covered with several reviews and comments, but should also include the Doody et al. *NEJM* 2013 study as a primary reference.
- The review by Haapasalo and Kovacs *JAD* 2011 lists 91 substrates and not more than 100 as stated in the manuscript. This should be corrected. However, it is clear gamma-secretase will likely have more than hundred substrates as the authors also state later in the manuscript.
- The primary citations for the S4 cleavage sites in Notch1 are missing.
- In the Discussion, the paper reporting three different gamma-secretase conformations should be cited (Bai et al. *eLife* 2015).
- The authors erroneously state in the Discussion that biomarkers other than A β do not exist. However, please note that the publication of Yanagida et al *EMM* 2009 proposed the APL1beta28 peptide as a surrogate marker for A β .
- Later in the discussion, also the key primary publications for the sequential cleavage of APP should be included.

Minor points:

- It should be explained what triple negative breast cancers and mammospheres are.
- For clarity, it should be stated that the Notch1-4 proteins studied are human as many previous studies have investigated mouse Notch1-4.
- on page 6, first line the term "gamma-secretase membranes" is not fully accurate as the enzyme was solubilized with CHAPSO.
- In the Material and Methods, the rationale for the K16A mutant in the Abeta epitope tag for the cell-based substrates should be given.
- In Fig. 8B, the same color code as in Fig. 8A should be used for the inhibitor data to avoid confusion.

Referee #3 (Remarks):

The manuscript by Ran and co-workers tackles the issue of using g-secretase inhibitors in cancer therapeutics given the facts that Notch is one of its most eye-catching substrates, Notch mutations are related to specific cancers and overall deregulated Notch signaling is found in selective cancers. Thus far, clinical trials addressing the therapeutic value of GSI in AD thus far failed and have diminished the enthusiasm for g-secretase as a valuable therapeutic target. However, much of the failure can be brought back to a genuine lack of an in-depth knowledge of the structure, molecular and pharmacological characteristics of g-secretase vis-à-vis its dozens of substrates. Likewise, the idea that many GSIs are pharmacological equivalent might be an oversimplification as well. With a renewed interest of using GSIs in cancer (but also other) therapy, this study now investigates several GSIs in the light of their effects on a broader range of bona fide substrates and extending these studies from several cancer cell lines to mammospheres. This study reveals for the first time the differential effects of GSIs on the processing of the different Notch substrates as well as on their role in inhibiting related signal peptide peptidases. As for APP they found effects on processivity and

potentiation of g-secretase cleavage in the different substrates. Overall this is a well-designed study that merits publication. On the other hand, the study is rather limited and maybe better suited as a short report, if the essential data and info can be adjusted to this format. I do have some inquiries that the authors should address.

Introduction:

p.4: Processing of Notch by g-secretase has been first described by De Strooper et al., Nature, 1999, instead of Saxena et al., 2001. In the same paragraph some refs should be added that refer to the different processing steps of Notch as well as the analogy with APP processing.

Results:

p.5, first paragraph: it is not clear from this first sentence how the fusion protein substrates are constructed. The authors refer to fig 1A but the reader has by him/herself to find out how it is composed. Moreover, this scheme differs from the scheme in fig 4A while I think this goes about the same construct: in other words, a signal peptide is missing in fig1A (and thus the model in fig4A becomes redundant).

Figure 2: Not all the arrows, depicting cleavage sites, are represented in the MS list/table. For instance for Notch1 I see more arrows (after each aa between aa1739-1746) than I see peaks in figure 1C. Can the authors explain how this summary was generated?

p.6: 'Following incubation with g-secretase membranes...' is a strange phrasing. Based on the description in M&M this appears to be a CHAPSO solubilized fraction and thus not a membrane fraction.

p.9, line 16: The authors refer to figure 6C stating that 'starting from 20nM, BMS, PF or RO significantly inhibited APP-CTFs'. They cannot argue for this as this is a single blot with no quantification and statistics. The authors should expand on these experiments to be able to provide statistics that support their claim.

Related to figure 7, the authors state that 'at 5µM, MK and PF, but none of the other GSIs, significantly reduced mammosphere numbers...'. Figure 7 shows that only PF-3084014 is significantly decreased, not MK-0752.

Discussion:

p.13: In the last paragraph the authors discuss the tri-peptide processing in the light of other substrates besides APP and that this general rule is not followed. However, in their recent paper Bolduc et al (2016) described the identification of similar three pocket binding sites in the catalytic region of g-secretase. Herein the authors suggest that a similar mechanism occurs for substrates like Notch which does not reconcile with the findings in the present manuscript. Can the authors discuss/explain this anomaly and include it in the discussion.

Textual errors:

P6, bottom: '... we nevertheless conducted several studies to examine...'

p.7, line 10: '... are detectable as well as potential dimers...'

p.7, line 20: 'Although the substrates produced (something is missing here?) Abeta-like peptides...'

1st Revision - authors' response

10 March 2017

Point by Point Response to the Reviewer's concerns.

We thank all three reviewers for their constructive critiques. We believe that we have addressed most of the concerns.

Referee #1 (Comments on Novelty/Model System):

Most of the technical data is sound but the cell biology assays for 'cancer stem cells' presented in figure 7 is technically and conceptually unsound and not fit for purpose.

Response: As noted below new data addressing this issue has been included.

Referee #1 (Remarks):

The paper is a much needed study of the diversity of gamma secretase inhibitor (GSI) action, with a particular focus on the inhibition of Notch receptor cleavage as a target in cancer. The details of the analyses of cleavage site preferences for individual gamma secretase targets are fascinating. In addition, the testing of the major chemical gamma secretase inhibitors developed the pharmaceutical

industry reveals distinct patterns of substrate specificity that impacts selectivity of Notch receptor inhibition. This is important and has relevance and impact for the potential of these in cancer treatment.

However, the investigation falters when it comes to testing of the cellular biological impact of these different specificities for cancer. The use of a single breast cancer cell line limits the investigation and the assay that purports to test an effect on cancer stem cells is not fit for purpose since it does not test colony formation or self-renewal. Many publications have described this assay and the key steps are to plate at clonal density, use the inhibitor at time 0 to test effects on stem/progenitor cells and replat without inhibitor to test for an effect on self-renewal. Such methods have been previously described in detail in the literature. None of the above methods are followed and the assay used where inhibitor is added after secondary plating is most likely to test for an effect on proliferation. Consequently, they observe little effect and little difference between inhibitors. This part would have to be strengthened considerably with use of additional cell lines and preferably an *in vivo* limiting dilution assay to have any confidence that an effect on cancer stem cells is being seen.

Response: We thank the review for the positive overall comments and the constructive criticisms. For limiting dilution assay we used two distinct TNBC cell lines. MDA-MB231 is the one we used before, and is molecularly "Mesenchymal", PTEN wild type. The other, MDA-MB468, is molecularly "Basal-like" and PTEN null. GSI treated mammospheres were dissociated and viable cells were re-plated at clonal density (1000, 100, and 10 cells) without inhibitor. At 5 μ M and especially at 10 μ M, all GSIs decreased mammosphere-forming ability. However, there were no significant differences among the tested GSIs in all conditions in both cell lines. These data suggest that inhibition of mammosphere forming ability is common to all GSIs we tested. However, when we examined absolute cell counts, the PF GSI was significantly more potent than the other compounds in both cell lines we studied at 5 and 10 μ M.

Next, we investigated whether PF GSI selectively affected cancer stem-like cells (CSCs) using both MDA-MB-231 and MDA-MB-468 cell lines. Mammospheres were treated with increasing concentration of PF GSI and CSCs (CD44⁺CD24^{low}) (Ref: EMBO Mol Med, 2013, 5, 1502-1522) were analyzed by Flow Cytometer. We found that PF GSI decreased the total cell number and the CSC fraction in both cell lines under these conditions, suggesting that it's not sparing "Stem-like" cells. The PF GSI caused a 15-20% relative decrease in CD44⁺CD24^{low} cells at all concentrations tested, and a dose-dependent decrease in the absolute numbers of CD44⁺CD24^{low} cells. Overall, these data suggest that the PF GSI has higher potency in mammosphere formation assays due to overall higher activity in both stem-like and non-stem like cells. Importantly, stem-like cells showed no relative resistance to this GSI, as they do to various cytotoxic chemotherapy agents. If anything, they appeared to be slightly more sensitive to this GSI than non-stem like cells.

Additional comments:

1) In the introduction and elsewhere, the citation of literature is limited too often to papers by the authors without reference to other relevant papers. For example, reference to decoys and antibodies for therapy fails to cite high impact papers (Eg. Kangsamaksin et al Cancer Discovery, 2015; Hoey et al., Cell Stem Cell, 2009)

Response: we have added those references.

2) Many acronyms are not defined on first use. Eg. APP, IB, CTP and NTP in abstract, results and figures.

Response: We have defined all abbreviations.

3) Fig. 4: For B, it is not clear which band the arrows are pointing to. In 4C and D, LY411575 is used but no structure is given in fig. 3 and it is not defined whether it is a GSI or not.

Response: We carefully adjusted the arrows to make sure they are pointed at the bands. LY411575 is a very potent GSI. We have added structure of LY411575 in to Fig 3 and defined it in the text.

4) Fig. 5: the line colours should be carefully chosen so that they can be distinguished.

Response: The color of RO4929079 and Semagacestat in Fig 5B were changed.

5) Page 8, lines 10-17: The effects shown in Figure 5 are very interesting and discussed here where an increase in cleavage at low doses of GSIs are seen, particularly affecting Notch3 and 4. This could explain the differential effects of DAPT and other GSIs on Notch1 versus Notch3/4 that have been previously published.

Response: Thanks for the comments. NOTCH3 and some of NOTCH4 cleavage showing a biphasic response in our cell free assay. There may be many explanations for this data, including stoichiometry between substrate; gamma-secretase complex and the drug play certain role in the low dose potentiation phenomenon.

6) Fig. 6: In A, the WB should indicate the band as N1-ICD. In B, the baseline control band intensity should be included. Error bars are shown but it is not clear from the legend whether these are from replicate WBs or not. Statistics and number of replicates should be included.

Response: We changed it to NOTCH1-ICD in A. In B we added the no EDTA group as the baseline. Error bars show standard error of three independent experiment. We also added relative band intensity of APP-CTF with 0.1uM GSI as new figure 6D. This information is now included in the Figure legend and methods.

7) I am unsure about the relevance of SPPLs as targets to the main purpose of the paper. It seems to be an add-on. Is it relevant?

Response: SPP/SPPLs inhibition profile shown in the manuscript just remind GSIs in clinical trial may have other effects that are "off-target". As we are one of the few labs that has these assays available, we think this data is important to include. Data suggest clinical GSIs have different inhibition potency on SPPLs. Indeed, SPP/SPPL have been implicated as important regulators of the immune system and likely have other roles that may be relevant for drugs being evaluated as cancer therapies.

8) The relevance of the mammosphere assay to cancer stem cells and effects of Notch inhibition should be properly discussed if suitable data addressing this are included.

Response: Based on additional data in Figure 7 data, we added to the discussion as suggested.

Referee #2 (Remarks):

In the present manuscript, Ran et al have investigated a variety of gamma-secretase inhibitors (GSIs) used in cancer clinical trials for their potencies and selectivity in inhibition of Notch and a few other substrates. As altered Notch signaling has been implicated in cancer, the focus of the study lies on the effects of the GSIs on the Notch1-4 substrates. In addition, the authors have investigated whether and how the selected GSIs would inhibit the gamma-secretase related SPP/SPPL proteases.

By identifying the gamma and epsilon cleavage sites in cell-free and cell-based assays, the authors show that Notch1-4, VEGFR and CD44 are processed in a similar way as APP. The authors further find that the GSIs tested can show differential effects on substrate cleavage and on cross-inhibition of SPP/SPPLs, such that the overall conclusion of the study is that the GSIs in cancer clinical trials are pharmacologically and functionally distinct.

As outlined below, I am afraid to say that the findings, conclusions and the concepts derived are, however, not really novel. It has been known since a long time that also other gamma-secretase substrates likely undergo sequential cleavages as APP by the identification of cleavage sites in the N-terminal to epsilon/site3 in a number of substrates (Notch1, CD44, APLP1, Neuregulin1). The authors confirm this concept by showing internal cleavages sites at gamma/site4 positions now also for Notch2, 3, and 4 and VEGFR. An interesting aspect is that some inhibitors seem to increase activity of processing at low-dose inhibition (e.g. for Notch3), a phenomenon which has already been known from APP processing.

In addition, that GSIs, which are not directed against the active such as the early transition-state analog inhibitors, which block cleavage of all substrates, can show differential effects on substrate cleavage with respect to inhibitor potency, is likely and has been shown earlier. Unlike the authors state in this study, I think it is commonly accepted in the field that GSIs are not necessarily

considered biological equivalents except those that directly target the active site. Other GSIs that not directly target the active site (and several if not all of the GSIs studied here likely fall into this category) can as mentioned above of course show a certain degree of substrate specificity. In fact, this concept has been the basis for the development of Notch-sparing GSIs for Alzheimer's disease therapy. Differential effects of GSIs and/or complex-specific inhibitors have been reported for PS1 and PS2 gamma-secretases by others, as well as for SPP/SPPLs by the authors earlier (Ran et al. PlosOne 2015).

Some inhibitors have slightly different potencies on Notch1 processing in breast cancer cell lines and some inhibitors inhibit mammosphere formation. Similar findings on mammosphere formation have already been reported for two other GSIs previously (Grudzien et al. Anticancer Res. 2010). These effects are interesting and working out to which substrates these relate could provide important novel insights. The implications and significance for cancer therapy, which these findings may have, do, however, not yet really become clear from this study.

Lastly, the technical quality of the data is overall very good, although some conclusions made also require further experimental substantiation. However, as said above, the manuscript provides only little advance over previous studies and the overall impact might therefore be only moderate.

Specific points:

Results:

- The cleavage sites and relative site usage (peak height) does not always match between both cell-free (Fig. 1C, e.g. for Notch1: A1741 is major peak) and cell-based (Fig. 4D, e.g. for Notch1: A1741 is minor peak) assays and also not with previous published studies (e.g. CD44 major gamma-cleavage site between LA and LA as reported by Lammich et al. JBC 2002, but in this study more N-terminal between A and S). These discrepancies may relate to different construct designs with respect to ectodomain length and N-terminal epitope-tag usage but would deserve a critical discussion.

Response: Thank you for the constructive comments. We have noticed the discrepancies between cell-free and cell-based assay. This has been noted in other published data. Different batches of substrate, reaction buffer preparations and stability of each peptides may all lead to difference of the peak intensities. The cleavage sites show more consistency but not the intensities.

- The lack of substrate accumulation shown in Fig. 4B is indeed very surprising. Is this also observed with other potent GSIs such as L-685,458? In clarifying this issue, the authors should also check CTF accumulation of a suitable endogenous substrate to assess whether the observed lack of substrate accumulation in the presence of GSIs may be due to substrate saturation upon overexpression. The immunoblots shown in Fig. 4B and Appendix Figure S4 are identical and not probed with different antibodies (Flag vs 6E10) as described (same main and background bands, same dirt, spots, etc.; please check carefully).

Response: We tried both of LY411575 (IC50=0.078 nM) and L685458 (IC50=17 nM) beside DAPT. In all cases only cAPPC100 show some accumulation just like in the figure. Substrate saturation upon overexpression may one reason, but other factors may contribute. For example, other clearance pathways may exist or the flux through gamma-secretase may be quite low for some substrates. We did find endogenous APP CTF accumulation with DAPT and other GSIs when using 231 cell (Fig 6), HEK and H4 cells. Though this is an intriguing issue, we do not think it alters the major conclusions and impact of the current findings.

Thanks the reviewer to point out the mistake of Fig 4B. We have change the Appendix Figure S4.

For Figure 5A-D, it would be desirable that these data are further substantiated by immunoblot analysis of NTF (Abeta-like peptide) and CTF (ICD) cleavage products to see the increase of activity at low-dose inhibition also in this experimental setting. This would probably best be done in the cell-free assay. With respect to the mass-spectrometry analysis shown in Fig. 5D for Notch1, in the accompanying Appendix Fig. 5B, the signals (peak heights) for Notch3 and Notch4 unexpectedly and strangely first go down, then up again and then down again. Again, immunoblot analysis of cleavage products would be needed to make these data more robust.

Response: The ELISA we use is incredibly well validated and has CV typically less than 10%. Western blotting of Abeta and Abeta like peptides is very challenging to use for quantification

especially in the presence of large amounts of recombinant substrate. Nevertheless, we put considerable effort into trying to use Western Blotting to confirm our data, but these did not produce data that was quantifiable.

Although unexpected when we reproducibly observe the signal of NOTCH3 and some of NOTCH4 cleavage showing a biphasic response, these data remain in the Appendix. There may be many explanations for this data, including stoichiometry between substrate; gamma-secretase complex and the drug play certain role in the low dose potentiation phenomenon. Again, these data do not alter the punch line of the manuscript, but we think are interesting points that the gamma-secretase aficionado's should find interesting.

- In Fig. 6, longer exposures comparable to that for BMS-906024 should be shown for the underexposed blots. Again, it appears that DAPT has not really been working (compare control 1 and lanes 4-6).

Response: At the given concentration of Semagacestat, MK-0752 and DAPT, we had never observed same CTF accumulation level comparable to BMS-906024. To repeat this experiment, we treated 231 cells with 100 nM of each GSI, all treatments were triplicated. We observed similar inhibition potencies. APP CTF accumulation with Semagacestat, effects MK-0752 and DAPT were hard to see. The data is now included in the new Fig. 6D.

Literature:

- The primary references for the finding that gamma-secretase is a complex composed of presenilin, nicastrin, APH-1 and PEN-2 are incomplete and should also include Edbauer et al. NatCellBiol 2003 and Kimberly et al. PNAS 2003.

Response: we added the references.

- The literature cited on the failure of GSI in AD clinical trials is covered with several reviews and comments, but should also include the Doody et al. NEJM 2013 study as a primary reference.

Response: we added the references.

- The review by Haapasalo and Kovacs JAD 2011 lists 91 substrates and not more than 100 as stated in the manuscript. This should be corrected. However, it is clear gamma-secretase will likely have more than hundred substrates as the authors also state later in the manuscript.

Response: we have corrected this.

- The primary citations for the S4 cleavage sites in Notch1 are missing.

Response: Mouse Notch1 cleavage site reference (Okochi et al JBC 2006) has been added into paragraph4 on page 7.

- In the Discussion, the paper reporting three different gamma-secretase conformations should be cited (Bai et al. eLife 2015).

Response: Yes, it was in the last paragraph of page12.

- The authors erroneously state in the Discussion that biomarkers other than A β do not exist. However, please note that the publication of Yanagida et al EMM 2009 proposed the APL1beta28 peptide as a surrogate marker for A β .

Response: We corrected this in the text. We were really referring to blood-based markers for Notch cleavages here, and were not precise enough in our writing. Thank you for calling this to our attention. We include this reference in our discussion.

-Later in the discussion, also the key primary publications for the sequential cleavage of APP should be included.

Response: Takami et al J Neurosci 2009 paper was added to the discussion.

Minor points:

- It should be explained what triple negative breast cancers and mammospheres are.

Response: We added explanation of triple negative breast cancer on page 5. Triple negative breast cancers are some type of breast cancer lack of estrogen receptor, progesterone receptor and human epidermal growth factor receptor2. Those three receptors present in most of other breast cancers. Mammosphere formation assay is widely used to identify sphere-forming cells that develop from stem-like cells.

- For clarity, it should be stated that the Notch1-4 proteins studied are human as many previous studies have investigated mouse Notch1-4.

Response: Thanks, we emphasized in the introduction, results and discussion that we are using human protein sequences.

- on page 6, first line the term "gamma-secretase membranes" is not fully accurate as the enzyme was solubilized with CHAPSO.

Response: Thanks, we corrected this.

- In the Material and Methods, the rationale for the K16A mutant in the Abeta epitope tag for the cell-based substrates should be given.

Response: The rationale is included (to decrease alpha-secretase cleavage), though we still saw Aβ1-15 and 1-16, as we have observed for unaltered APP C99 substrates. Thus, we likely could have used a wild type APP tag. Again, this does really change any interpretation of our data. The rationale was included in the methods section describing these constructs.

- In Fig. 8B, the same color code as in Fig. 8A should be used for the inhibitor data to avoid confusion.

Response: We have corrected this.

Referee #3 (Remarks):

The manuscript by Ran and co-workers tackles the issue of using g-secretase inhibitors in cancer therapeutics given the facts that Notch is one of its most eye-catching substrates, Notch mutations are related to specific cancers and overall deregulated Notch signaling is found in selective cancers. Thus far, clinical trials addressing the therapeutic value of GSI in AD thus far failed and have diminished the enthusiasm for g-secretase as a valuable therapeutic target. However, much of the failure can be brought back to a genuine lack of an in-depth knowledge of the structure, molecular and pharmacological characteristics of g-secretase vis-à-vis its dozens of substrates. Likewise, the idea that many GSIs are pharmacological equivalent might be an oversimplification as well. With a renewed interest of using GSIs in cancer (but also other) therapy, this study now investigates several GSIs in the light of their effects on a broader range of bona fide substrates and extending these studies from several cancer cell lines to mammospheres. This study reveals for the first time the differential effects of GSIs on the processing of the different Notch substrates as well as on their role in inhibiting related signal peptide peptidases. As for APP they found effects on processivity and potentiation of g-secretase cleavage in the different substrates. Overall this is a well-designed study that merits publication. On the other hand, the study is rather limited and maybe better suited as a short report, if the essential data and info can be adjusted to this format. I do have some inquiries that the authors should address.

Introduction:

p.4: Processing of Notch by g-secretase has been first described by De Strooper et al., Nature, 1999, instead of Saxena et al., 2001. In the same paragraph some refs should be added that refer to the different processing steps of Notch as well as the analogy with APP processing.

Response: Thanks, we added De Strooper et al. Nature 1999, Okochi et al. JBC 2006 and De Strooper et al. Nat Rev Neurol 2010.

Results:

p.5, first paragraph: it is not clear from this first sentence how the fusion protein substrates are constructed. The authors refer to fig 1A but the reader has by him/herself to find out how it is composed. Moreover, this scheme differs from the scheme in fig 4A while I think this goes about the same construct: in other words, a signal peptide is missing in fig1A (and thus the model in fig4A becomes redundant).

Response: We have now clarified this potentially confusing detail. We have described the constructs in detail in the Material and Methods. Fig1A is the scheme for E. Coli overexpression so no signal peptide was used in the constructs. Fig4A is for mammalian cell culture and signal peptide was included. They are also different in the length of A β peptide. The E coli construct has A β 1-15, the mammalian construct has A β 1-25 for ELISA using 4G8 antibody.

Figure 2: Not all the arrows, depicting cleavage sites, are represented in the MS list/table. For instance for Notch1 I see more arrows (after each aa between aa1739-1746) than I see peaks in figure 1C. Can the authors explain how this summary was generated?

Response: We again have clarified this in the text and figure legend. Figure 2 and Table 1 are summaries of the results from cell-free AND cell-based assays. Some peaks are only present in one of those two assays. For example, V1739 site was only found in cell-based assay. Some minor cleavage sites listed in Table 1 were not labeled in the MS figure.

p.6: 'Following incubation with g-secretase membranes...' is a strange phrasing. Based on the description in M&M this appears to be a CHAPSO solubilized fraction and thus not a membrane fraction.

Response: We have altered the text.

p.9, line 16: The authors refer to figure 6C stating that 'starting from 20nM, BMS, PF or RO significantly inhibited APP-CTFs'. They cannot argue for this as this is a single blot with no quantification and statistics. The authors should expand on these experiments to be able to provide statistics that support their claim.

Response: For better quantification, we treated 231 cells with 100 nM of each GSI. All tests were triplicate. The results were shown in the new Figure 6D. APP CTF increased >15 times when treated with BMS, RO and PF compounds. Semagacestat, MK and DAPT compounds did not significantly increase CTF level.

Related to figure 7, the authors state that 'at 5 μ M, MK and PF, but none of the other GSIs, significantly reduced mammosphere numbers...'. Figure 7 shows that only PF-3084014 is significantly decreased, not MK-0752.

Response: We re-designed and performed the mammospheres experiment using limiting dilution assay (new Figure 7). We also performed the assay on another TNBC cell line MDA-MB-468. New results suggested PF-3084014 is more potent than the other GSIs. (see response to first reviewer above)

Discussion:

p.13: In the last paragraph the authors discuss the tri-peptide processing in the light of other substrates besides APP and that this general rule is not followed. However, in their recent paper Bolduc et al (2016) described the identification of similar three pocket binding sites in the catalytic region of g-secretase. Herein the authors suggest that a similar mechanism occurs for substrates like Notch which does not reconcile with the findings in the present manuscript. Can the authors discuss/explain this anomaly and include it in the discussion.

Response. We have included the Bolduc reference and several others that demonstrate that simply more work is needed here to resolve how each substrate is processed sequentially. Again this is probably a point more for the hard-core enzymologists and does not alter the main point of this current study. The revised reads as follow:

" At present, it is hard to see how a consistent, processive cleavage model involving only tri or tetrapeptides, could account for the cleavage patterns we have observed in this study. Although this model has been invoked, and to some extent experimentally supported, with respect to cleavage of APP (Bolduc et al, 2016; Takami et al, 2009), other studies demonstrate that initial γ -secretase cleavage does not precisely define subsequent product lines (Ran et al, 2014) and that penta- and hexapeptides can be released during processing of APP by γ -secretase (Matsumura et al, 2014). Additional studies will be needed to understand how for example, a single initial endopeptidase cleavage of NOTCH1 and 2 substrates eventually produces 9 A β -like peptides with no consistent spacing between the final cleavage sites. In these instances a tripeptide step-wise cleavage model cannot account for the complexity of final products that are detected."

Textual errors:

P6, bottom: '... we nevertheless conducted several studies to examine....'

Response: Corrected.

p.7, line 10: '... are detectable as well as potential dimers...'

Response: we corrected this.

p.7, line 20: 'Although the substrates produced (something is missing here?) Abeta-like peptides....'

Response: change to 'Although A β -like peptides are detectable by ELISA, we were not able to detect cleavage product of cNOTCH4sub, cCD44sub and cVEGFR1sub in the overexpressing cell lines using IP-MS'

2nd Editorial Decision

11 April 2017

Thank you for the submission of your revised manuscript to EMBO Molecular Medicine. We have now received the enclosed reports from the referees that were asked to re-assess it. As you will see the reviewers are now globally supportive and I am pleased to inform you that we will be able to accept your manuscript pending the following final amendments:

1) Please address referee 1's comments experimentally as much as possible. Please provide a letter INCLUDING the reviewer's reports and your detailed responses to their comments (as Word file).

Please submit your revised manuscript within two weeks. I look forward to seeing a revised form of your manuscript as soon as possible.

***** Reviewer's comments *****

Referee #1 (Comments on Novelty/Model System):

The method for analysis of the CD44+CD24low population by FACS needs revision.

Referee #1 (Remarks):

I am happy that the authors have improved the analysis of mammosphere formation after GSI treatments and extended it to an additional cell line. In figure 7C, they include analysis of a cancer stem-like population which are CD44+/CD24low by FACS and cite Azzam et al 2013 for this method. However, I find that they do not follow this method since Azzam et al found approximately 15% of cells have this marker in 231 cells while the majority of cells were CD44+/CD24- and were not stem cell-like. In figure 7C, the authors report ~90% of 231 and 468 cells are CD44+/CD24low. Taken at face value, this would indicate that the cell lines used are comprised largely of stem-like cells. It seems likely that they have merged the CD44+ populations that are CD24-negative and low,

where these populations should be analysed separately. This error should be corrected before it is acceptable for publication.

Referee #2 (Remarks):

The authors have addressed my previously raised specific points as far as possible. However, the major general concerns, outlined in detail in my previous review, regarding the limited novelty did not really change with this revised version. In my overall assessment, the revised manuscript does still not provide much advance over previous studies.

Comments:

In the new experiments requested by reviewer 1, the inhibitors tested behaved overall very similar on inhibition of mammosphere formation in the limited dilution assay with one compound (PF-3084014) being somewhat more potent than all others. This new finding may challenge their conclusion that such GSIs are functionally distinct, at least there is not much indication of this in the new assays requested by reviewer 1 (Fig. 7). How the different inhibition potencies of the GSIs determined in cell-free and cell-based assays impact on γ -secretase inhibition in cancer cells has as a starting point so far only been assessed for Notch1 (Fig. 6). How they would impact for comparison e.g. on Notch 2-4 cleavage in such cells remain important open questions directly related to this study. Such studies would provide more insight into GSIs used in cancer trials and could really provide an "important framework" for the evaluation of data coming from such cancer trials. With the present data shown, this goal is not yet reached.

Referee #3 (Remarks):

The authors have addressed all my concerns.

2nd Revision - authors' response

24 April 2017

***** Reviewer's comments *****

Referee #1 (Comments on Novelty/Model System):

The method for analysis of the CD44+CD24^{low} population by FACS needs revision.

Referee #1 (Remarks):

I am happy that the authors have improved the analysis of mammosphere formation after GSI treatments and extended it to an additional cell line. In figure 7C, they include analysis of a cancer stem-like population which are CD44+/CD24^{low} by FACS and cite Azzam et al 2013 for this method. However, I find that they do not follow this method since Azzam et al found approximately 15% of cells have this marker in 231 cells while the majority of cells were CD44+/CD24⁻ and were not stem cell-like. In figure 7C, the authors report ~90% of 231 and 468 cells are CD44+/CD24^{low}. Taken at face value, this would indicate that the cell lines used are comprised largely of stem-like cells. It seems likely that they have merged the CD44⁺ populations that are CD24⁻ and low, where these populations should be analysed separately. This error should be corrected before it is acceptable for publication.

Response: We thank the reviewer for pointing this out. Flow data were re-analyzed using the Kaluza Analysis software (Beckman Coulter), which gave us better density plots compared to data from the resident software in our flow cytometer. CD44+CD24^{low} cells are 13-20%, comparable with Azzam et al. We revised Figure 7C with the new analysis results. We also changed the text on Page 10 to: "PF-3084014 caused a decrease in percentage of CD44+CD24^{low} cells at all concentrations tested,"

Referee #2 (Remarks):

The authors have addressed my previously raised specific points as far as possible. However, the major general concerns, outlined in detail in my previous review, regarding the limited novelty did not really change with this revised version. In my overall assessment, the revised manuscript does still not provide much advance over previous studies.

Comments:

In the new experiments requested by reviewer 1, the inhibitors tested behaved overall very similar on inhibition of mammosphere formation in the limited dilution assay with one compound (PF-3084014) being somewhat more potent than all others. This new finding may challenge their conclusion that such GSIs are functionally distinct, at least there is not much indication of this in the new assays requested by reviewer 1 (Fig. 7). How the different inhibition potencies of the GSIs determined in cell-free and cell-based assays impact on γ -secretase inhibition in cancer cells has as a starting point so far only been assessed for Notch1 (Fig. 6). How they would impact for comparison e.g. on Notch 2-4 cleavage in such cells remain important open questions directly related to this study. Such studies would provide more insight into GSIs used in cancer trials and could really provide an "important framework" for the evaluation of data coming from such cancer trials. With the present data shown, this goal is not yet reached.

Response: We agree with the reviewer that examining multiple substrates and multiple Notch paralogs is essential to comparing different GSIs. Unfortunately, Western blotting is not ideal for this purpose, due to the lack of reliable antibodies specific for the cleaved epitopes of other paralogs. With commercially available antibodies to Notch2-4, which we have tested, obtaining quantitative results and even clearly identifying the NICD bands without mass spectrometry validation is difficult. For this reason, we developed the mini-gene based artificial substrates described in Figure 1 and 4. In these assays, PF-3084014 is considerably more potent against Notch2 than other GSIs (Figure 5A).

Referee #3 (Remarks):

The authors have addressed all my concerns.

Corresponding Author Name: Todd Golde
Journal Submitted to: EMBO Molecular Medicine
Manuscript Number: 1